# CFASL: Composite Factor-Aligned Symmetry Learning for Disentanglement in Variational AutoEncoder

## Abstract

Implemented symmetries of input and latent vectors is important for disentanglement learning in VAEs, but most works focus on disentangling each factor without consideration of multi-factor change close to real world transformation between two samples, and even a few studies to handle it in autoencoder literature are constrained to pre-defined factors. We propose a novel disentanglement framework for Composite Factor-Aligned Symmetry Learning (CFASL) on VAEs for the extension to general multi-factor change condition without constraint. CFASL disentangles representations by 1) aligning their changes, explicit symmetries, and unknown factors via proposed inductive bias, 2) building a composite symmetry for multi-factor change between two samples, and 3) inducing group equivariant encoder and decoder in the condition. To set up the multi-factor change condition, we propose sample pairing for inputs, and an extended evaluation metric. In quantitative and in-depth qualitative analysis, CFASL shows significant improvement of disentanglement in multi-factor change condition compared to state-of-the-art methods and also gradually improves in single factor change condition on common benchmarks.

## 1 Introduction

Disentangling representations by intrinsic factors of datasets is a crucial issue in machine learning literature (Bengio et al., 2013). In Variational Autoencoder (VAE) frameworks, a widely-used method to handle the issue is to factorize latent vector dimensions to contain specific factor information (Kingma & Welling, 2013; Higgins et al., 2017; Chen et al., 2018; Kim & Mnih, 2018; Jeong & Song, 2019; Shao et al., 2020; 2022). Although their effective disentanglement learning methods, Locatello et al. (2019) raises the serious difficulty of disentanglement without sufficient inductive bias.

In VAE literature, recent works using group theory provide a possible solution to inject such inductive bias by decomposing group symmetries (Higgins et al., 2018) in the latent vector space. To implement group equivariant VAE, Winter et al. (2022); Nasiri & Bepler (2022) achieve the translation and rotation equivariant VAE. The other branch implements the group equivariant function (Yang et al., 2022; Keller & Welling, 2021b) over the pre-defined group actions. All of the methods effectively enhance disentanglement by adjusting symmetries, but they focused on symmetry control in only single factor change or simple multi-factor change condition rather than unconstrained multi-factor change.

Multi-factor change is important for symmetry control because transformation between two input samples is unrestricted to single factor change in the real world. This issue has rarely been raised in VAE frameworks. In other literature on autoencoder to control symmetries (Miyato et al., 2022; Bouchacourt et al., 2021; Guo et al., 2019; Quessard et al., 2020; Shakerinava et al., 2022), there also exist a few recent works to consider composite symmetry for the multi-factor change, but they are constrained by allowing weak supervision to use factor class information of input pairs (Marchetti et al., 2023). Furthermore, the methods on autoencoder are not directly applicable to VAEs, because of the large difference to VAE in probabilistic interpretation.

In this paper, we propose a novel disentanglement method for Composite Factor-Aligned Symmetry Learning (CFASL) on VAE frameworks to address the multi-factor change condition via the following

distinguished approaches: 1) network architecture to learn an explicit codebook of symmetries responsible for each single factor change, called *factor-aligned* symmetries, and their composition for representing multi-factor change, 2) training losses to inject inductive bias for disentanglement via interpreting representation changes as the explicit symmetries and directly adjusting their properties, 3) implementing group equivariant encoder and decoder functions for disentanglement in the multi-factor change condition, 4) a problem setting that uses a pair of samples as an input without any information of factor labels, and 5) an extended metric (m-FVM$_k$) to evaluate disentanglement in the multi-factor change condition. We quantitatively and qualitatively analyze the method in common benchmarks of disentanglement on VAEs.

## 2 DIFFICULTY OF DISENTANGLING REPRESENTATIONS IN MULTI-FACTOR CHANGE

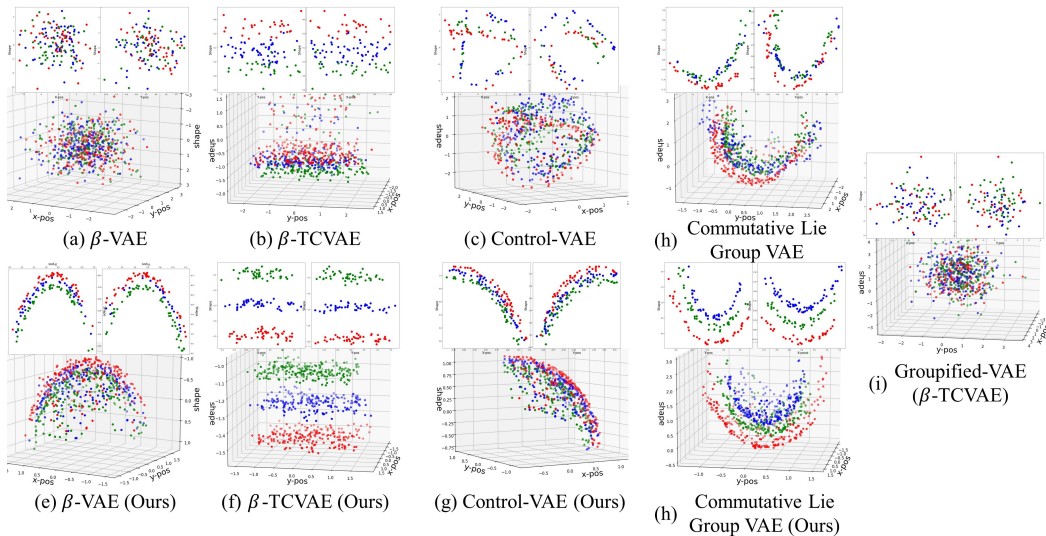

Figure 1: Distribution of latent vectors for dimensions responsible for Shape, X-pos, Y-pos factors in the dSprites dataset (Matthey et al., 2017). The groupified-VAE method is applied to $\beta$-TCVAE because this model shows a better evaluation score (Yang et al., 2022), and its reproducibility is introduced in Appendix D.2. The results show disentanglement for Shape from the combination of the other two factors by coloring three shapes (square, ellipse, heart) as red, blue, and green color, respectively. Each 3D-plot shows the whole distribution, and 2D-plots are cross-sections passing the center area of the grid. The Left and right 2D-plot are perpendicular to X-pos and Y-pos axis, respectively. We fix Scale and Orientation factor values, and plot randomly sampled 640 inputs (20.8% of all possible observation ($32 \times 32 \times 3 = 3,072$)). We select the dimensions responsible for the factors by selecting the largest value of the Kullback-Leibuler divergence between the prior and the posterior.

In this paper, we define *multi-factor change* condition as simultaneously altering more than two factors in the transformation between two samples or representations. In current disentanglement methods (Miyato et al., 2022; Bouchacourt et al., 2021; Guo et al., 2019; Quessard et al., 2020; Shakerinava et al., 2022), this issue has not been directly considered in implementation (Zhu et al., 2021), or has been handled in a limited environment of fixing specific pre-defined symmetries (Yang et al., 2022).

To confirm the impact of the issue, we tested them in disentangling of Shape factor from the combination of X-position and Y-position factors. In the first row of Fig. 1, the change of the dimension value corresponding to Shape is not consistently adjusted by the change of latent dimension mostly used for the factor, when the x-pos and y-pos change. This phenomenon implies that disentangled representation in multi-factor change is still insufficiently handled. Our approach is to directly consider the multi-factor change condition without factor limitation in VAE frameworks as the result in the second row of Fig. 1.

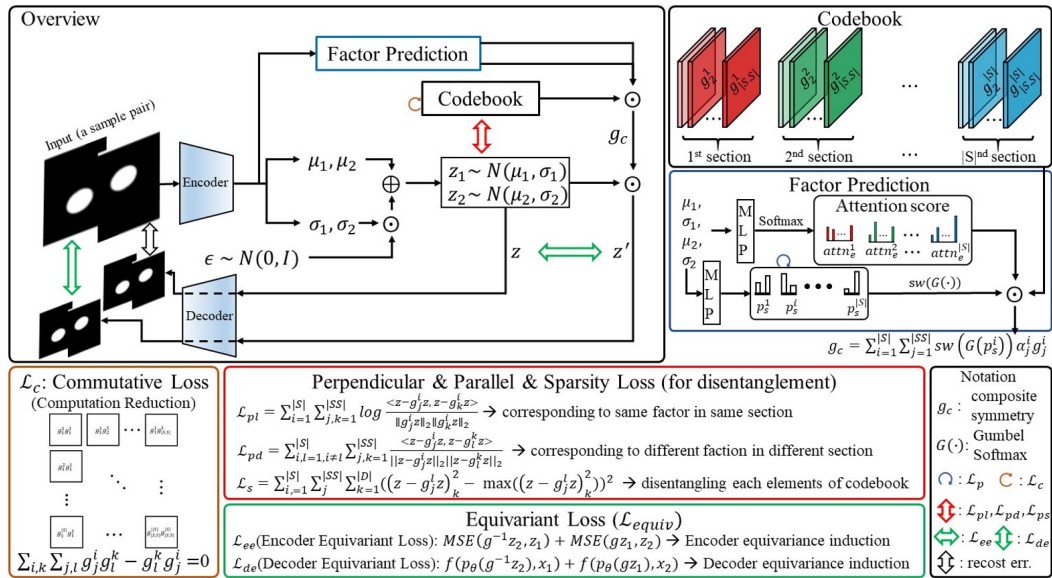

Figure 2: The overall architecture of our proposed method. The loss function is divided into four parts: 1) commutative loss ($\mathcal{L}_c$), 2) perpendicular, parallel, and sparsity loss ($\mathcal{L}_{pd}$, $\mathcal{L}_{pl}$, and $\mathcal{L}_s$) in Equation 2-4, 3) factor prediction loss ($\mathcal{L}_p$) in Equation 5), and 4) equivariant loss ($\mathcal{L}_{ee}$, and $\mathcal{L}_{de}$)in Equation 9). MLP is a multi-layer perceptron, and, $tr$ is a threshold. Attention score $attn_e$, $sw(\cdot)$, and $p_s^i$ are introduced in section 3.3.

## 3 METHODS

### 3.1 PROBLEM SET-UP FOR LEARNING FROM MULTI-FACTOR CHANGE

**Input and Output: A Pair of Two Samples** To help a VAE recognize a multi-factor change of two real samples and learn symmetries for the change directly, we give a pair of two different samples as an input. During the training, samples in the mini-batch $\mathbb{X}_{|B|}$ are randomly divided into two parts $\mathbb{X}_{|B|}^1 = \{x_1^1, x_2^1, \ldots, x_{\frac{|B|}{2}}^1\}$, and $\mathbb{X}_{|B|}^2 = \{x_1^2, x_2^2, \ldots, x_{\frac{|B|}{2}}^2\}$, where $|B|$ is a mini-batch size. In the next, our model pairs the samples $(x_1^1, x_1^2), (x_2^1, x_2^2), \ldots, (x_{\frac{|B|}{2}}^1, x_{\frac{|B|}{2}}^2)$ and is used for learning symmetries between the elements of each pair.

**Evaluation: m-FVM Metric for Disentanglement in Multi-Factor Change** As far as we investigated, there is no evaluation metric for disentanglement in multi-factor change, so we propose the extended version of the Factor-VAE metric (FVM) score called as multi-FVM score (m-FVM$_k$), where $k \in \{2, 3, \ldots, |F|-1\}$, and $|F|$ is a number of factors. Similar to FVM, 1) we randomly choose the $k$ fixed factors $(F_i, F_j, \ldots)$, 2) sample each factor's value $(f_i, f_j, \ldots)$ and fix the corresponding factor dimension value in the mini-batch, where $f_i \in \{1, 2, \ldots |F_i|\}, f_j \in \{1, 2, \ldots |F_j|\}, \ldots, |F_i|$ and $|F_j|$ is a maximum value of each factor label. 3) Then we estimate the standard deviation (std.) of each dimension to find the number of $k$ lowest std. dimension $(z_{l1}, z_{l2}, \ldots)$ in one epoch. In the next, 4) counting each pair of selected dimensions by std. (the number of $(z_{l1}, z_{l2}, \ldots)$, which are corresponded to fixed factors) and 5) add the maximum value of the number of $(z_{l1}, z_{l2}, \ldots)$ on all fixed factor cases, and divide with epoch.

**Objective and Base model** Our method can be plugged into existing VAE frameworks, so the objective function is additively integrated as

$$\mathcal{L}(\phi, \theta; \boldsymbol{x}) = \mathcal{L}_{VAE} + \mathcal{L}_{codebook} + \mathcal{L}_{equiv}, \tag{1}$$

where $\mathcal{L}_{VAE}$ is the loss function of a VAE framework (Appendix A). The other loss $\mathcal{L}_{codebook} = \mathcal{L}_{pl} + \mathcal{L}_{pd} + \mathcal{L}_s + \mathcal{L}_c + \mathcal{L}_p$ and $\mathcal{L}_{equiv} = \mathcal{L}_{ee} + \epsilon \mathcal{L}_{de}$ where $\epsilon$ is a hyper-parameter, which are introduced in the following subsections.

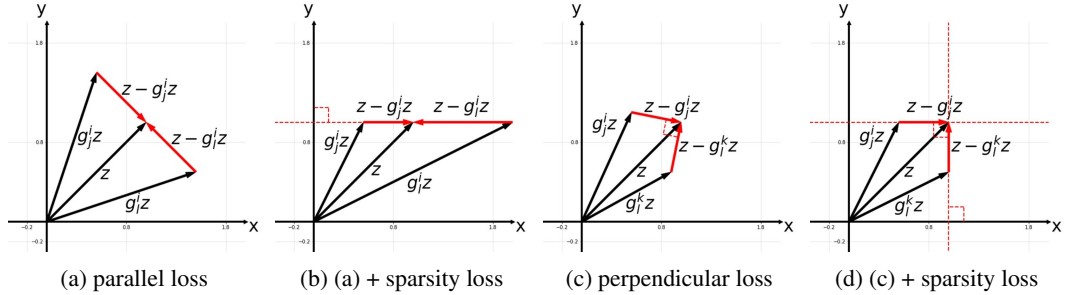

(a) parallel loss   (b) (a) + sparsity loss   (c) perpendicular loss   (d) (c) + sparsity loss

Figure 3: Roles of parallel, perpendicular, and sparsity loss on symmetries in the codebook for adjusting representation change. Parallel loss is for symmetries of the same section, and perpendicular loss is for different sections. Each axis (x and y) only affects to a single factor.

## 3.2 FACTOR-ALIGNED SYMMETRY LEARNING WITH INDUCTIVE BIAS

On the symmetry codebook, we inject inductive bias for disentanglement via Parallel loss $\mathcal{L}_{pl}$ and Perpendicular loss $\mathcal{L}_{pd}$ that adjust relations between latent vector changes by the symmetries. To align the changes for a factor to a dimension of latent vector $z$, we add sparsity loss $\mathcal{L}_s$ to the losses as shown in Fig. 3. Also, we implement the commutative loss $\mathcal{L}_c$ to reduce the computational costs for matrix exponential multiplication.

**Explicit and Learnable Symmetry Representation for Inductive Bias Injection** To allow direct injection of inductive bias to symmetries, we implement an explicit and learnable symmetry codebook. The codebook $\mathcal{G} = \{\mathcal{G}^1, \mathcal{G}^2, \ldots, \mathcal{G}^k\}$ is to assign a section $\mathcal{G}^i$ to a single factor, where $k \in \{1, 2, \ldots |S|\}$, and $|S|$ is the number of sections. The section $\mathcal{G}^i$ is composed of Lie algebra $\{\mathfrak{g}_1^i, \mathfrak{g}_2^i, \ldots, \mathfrak{g}_l^i\}$, where $\mathfrak{g}_j^i \in \mathbb{R}^{|D| \times |D|}$, $l \in \{1, 2, \ldots, |SS|\}$, $|SS|$ is the number of elements in each section, and $|D|$ is a dimension size of latent $z$. We assume that each Lie algebra consists of linearly independent bases $\mathfrak{B} = \{\mathfrak{B}_i | \mathfrak{B}_i \in \mathbb{R}^{n \times n}, \sum_i \alpha_i \mathfrak{B}_i \neq 0, \alpha_i \neq 0\}$: $\mathfrak{g}_j^i = \sum_b \alpha_b^{i,j} \mathfrak{B}_b$, where $b \in \{1, 2, \ldots, kl\}$. Then the dimension of the element of the codebook is equal to $|\mathfrak{B}|$ and the dimension of the Lie group which is composited by the codebook element is also $|\mathfrak{B}|$. To utilize previously studied effective expression of symmetry for disentanglement, we set the symmetry to be continuous (Higgins et al., 2022) and invertible via matrix exponential form (Xiao & Liu, 2020) as $g_j^i = \mathbf{e}^{\mathfrak{g}_j^i} = \sum_{k=0}^{\infty} \frac{1}{k!} (\mathfrak{g}_j^i)^k$ to construct the Lie group (Hall, 2015).

**Inductive Bias: Parallel Change by Symmetries of A Same Factor** We add a bias that latent vector changes by two symmetries for the same factor should be parallel ($z - g_j^i z \parallel z - g_l^i z$ for $i$th section) as shown in Fig. 3a. We define a loss function to make them parallel as:

$$\mathcal{L}_{pl} = \sum_{i=1}^{|S|} \sum_{j,k=1}^{|SS|} \log \frac{< z - g_j^i z, z - g_k^i z >}{||z - g_j^i z||_2 \cdot ||z - g_k^i z||_2}, \tag{2}$$

where $g_j^i = \mathbf{e}^{\mathfrak{g}_j^i}$, $< \cdot, \cdot >$ is a dot product, and $|| \cdot ||_2$ is a L2 norm.

**Inductive Bias: Perpendicular Change by Symmetries of Different Factors** Similarly to the parallel loss, we inject another bias that changes by two symmetries for different factors should satisfy the orthogonality ( $z - g_j^i z \perp z - g_l^k z$ for different $i$th and $k$th sections) as shown in Fig. 3c. The loss for inducing the orthogonality is

$$\mathcal{L}_{pd} = \sum_{i,k=1, i \neq k}^{|S|} \sum_{j,l=1}^{|SS|} \frac{< z - g_j^i z, z - g_l^k z >}{||z - g_j^i z||_2 \cdot ||z - g_l^k z||_2}. \tag{3}$$

This loss is computationally expensive to calculate ($O(|S|^2 \cdot |SS|^2)$), so we randomly select a $(j, l)$ pair of symmetries of different sections. This random selection still holds the orthogonality, because if all elements in the same section satisfy Equation 2 and a pair of elements from a different section ($\mathcal{G}^i, \mathcal{G}^j$) satisfies Equation 3, then any pair of the element ($\mathfrak{g}^i \in \mathcal{G}^i, \mathfrak{g}^j \in \mathcal{G}^j$) satisfies the Equation 3. More details are in Appendix B.

**Inductive Bias: Entangling A Factor to Only A Latent Dimension**    Factorizing latent dimensions to represent the change of separate factors is an attribute of disentanglement defined in Bengio et al. (2013) and derived by ELBO term in VAE training frameworks (Chen et al., 2018; Kim & Mnih, 2018). However, the parallel and perpendicular loss has no specific constraint to generate the attribute as shown in Fig. 3a, 3c. To guide symmetries to hold the attribute, we enforce the change $\Delta_j^i \boldsymbol{z} = \boldsymbol{z} - g_j^i \boldsymbol{z}$ to be a parallel shift to a unit vector as Fig. 3b, 3d via sparsity loss defined as

$$\mathcal{L}_s = \sum_{i=1}^{|S|} \sum_{j}^{|SS|} \Big[ \big[ \sum_{k=1}^{|D|} (\Delta_j^i \boldsymbol{z}_k)^2 \big]^2 - \max_k ((\Delta_j^i \boldsymbol{z}_k)^2)^2 \Big], \text{ where } \Delta_j^i \boldsymbol{z}_k \text{ is a } k^{th} \text{ dimension value.} \quad (4)$$

### 3.3    Composition of Factor-Aligned Symmetries via Two-Step Attention

**First Step: Elements of Section Selection**    In the first step, the model chooses the appropriate symmetries of each section through the attention score: $attn_e^i = softmax([M;\Sigma]\boldsymbol{W}_c^i + \boldsymbol{b}_c^i)$, where $[M;\Sigma] = [\boldsymbol{\mu}_1; \boldsymbol{\sigma}_1; \boldsymbol{\mu}_2; \boldsymbol{\sigma}_2]$, $\boldsymbol{W}_c^i \in \mathbb{R}^{4|D| \times |SS|}$ and $\boldsymbol{b}_c^i \in \mathbb{R}^{|SS|}$ are learnable parameters, and $i \in \{1, 2, \ldots |S|\}$.

**Second Step: Section Selection**    In the second step, the proposed model forces to predict the factors that we consider to have changed. We assume that if some factor value of two inputs is equal, then the variance of the corresponding latent vectors dimension value is smaller than others. By this assumption, we set the target for factor prediction: if $\boldsymbol{z}_{1,i} - \boldsymbol{z}_{2,i} >$ threshold, then we set $T_i$ as 1 and 0 otherwise, where $T_i$ is a $i^{th}$ dimension value of $T \in \mathbb{R}^{|D|}$, $\boldsymbol{z}_{j,i}$ is an $i^{th}$ dimension value of $\boldsymbol{z}_j$, and we set the threshold as a hyper-parameter. For section prediction, we utilize the cross-entropy loss:

$$\mathcal{L}_p = \sum_{i=1}^{|S|} \sum_{c \in C} \mathbb{1}[T_i = c] \cdot \log softmax(p_s^i), \text{ where } p_s^i = [M;\Sigma]\boldsymbol{W}_s^i + \boldsymbol{b}_s^i, \quad (5)$$

$\boldsymbol{W}_s^i \in \mathbb{R}^{4|D| \times 2}$ and $\boldsymbol{b}_s^i \in \mathbb{R}^2$ are learnable parameters, and $c \in \{0, 1\}$.

To infer the activated section of the symmetries codebook, we utilize the Gumbel softmax function to implement on-and-off cases like a switch:

$$sw(G(p_s^i)) = \begin{cases} G(p_{s,2}^i) & \text{if } p_{s,2}^i \geq 0.5 \\ 1 - G(p_{s,1}^i) & \text{if } p_{s,2}^i < 0.5 \end{cases}, \quad (6)$$

where $p_{s,j}^i$ is a $j^{th}$ dimension value of $p_s^i$, and $G(\cdot)$ is the Gumbel softmax with temperature as 1e-4.

**Integration for Composite Symmetry**    For the composite symmetry $g_c$, we apply the weighted sum of switch function $sw(p_s)$ and prediction distribution $attn_e$ as: $g_c = \prod_{i=1}^{|S|} \prod_{j=1}^{|SS|} g_j^i$, where $g_j^i = \mathrm{e}^{sw(G(p_s^i)) \cdot attn_{e,j}^i \cdot \mathfrak{g}_i^j}$, and $attn_{e,j}^i$ is a $j^{th}$ dimension value of $attn_e^i$.

**Commutativity Loss for Computational Efficiency**    In the composite symmetry $g_c$ calculation, the production $\prod_{i=1}^{|S|} \prod_{j=1}^{|SS|} g_j^i$ is a computationally expensive Taylor series repeated for all $(i,j)$ pairs. To reduce the cost by repetition, we enforce all pairs of basis $\mathfrak{g}_i^j$ to be commutative to convert the production to $\mathrm{e}^{\sum_{i,j} g_j^i}$ (By the matrix exponential property: $\mathrm{e}^{\boldsymbol{A}}\mathrm{e}^{\boldsymbol{B}} = \mathrm{e}^{\boldsymbol{A}+\boldsymbol{B}}$ as $\boldsymbol{AB} = \boldsymbol{BA}$, where $\boldsymbol{A}, \boldsymbol{B} \in \mathbb{R}^{n \times n}$). The loss for the commutativity is $\mathcal{L}_c = \sum_{i,k=1}^{|S|} \sum_{j,l=1}^{|SS|} \mathfrak{g}_j^i \mathfrak{g}_l^k - \mathfrak{g}_l^k \mathfrak{g}_j^i \to \boldsymbol{0}$.

### 3.4    Equvariance Induction of Composite Symmetries

**How to Induce Equivariance?**    Motivated by the equivariant mapping implementations (Yang et al., 2022; Miyato et al., 2022) for disentanglement, we implement an equivariant encoder and decoder that satisfies $q_\phi(\psi_i * x) = g_i \circ q_\phi(x)$ and $p_\theta(g_i \circ z) = \psi_i * p_\theta(z)$ respectively, where $q_\phi$ is an encoder, and $p_\theta$ is the decoder. In the notation, $\psi_i$ and $g_i$ are group elements of the group $(\Psi, *)$ and $(\mathcal{G}, \circ)$ respectively, and both groups are isomorphic. Each group acts on the input and latent vector space with group action $*$, and $\circ$, respectively. We specify the form of symmetry $g_i$, and $\circ$ as an invertible matrix, and group action as matrix multiplication on the latent vector space. Then, the

encoder equivariant function can be rewritten by multiplying the inversion of $g_i$ on both sides and $z$ can be replaced with the $q_\phi(x)$ in the decoder equivariant function as

$$q_\phi(x) = g_i^{-1} \circ q_\phi(\psi_i * x) \iff q_\phi(x) - g_i^{-1} \circ q_\phi(\psi_i * x) \to 0 \qquad \text{(for encoder)} \quad (7)$$

$$p_\theta(g_i \circ q_\phi(x)) = \psi_i * p_\theta(q_\phi(x)) \iff p_\theta(g_i \circ q_\phi(x)) - \psi_i * p_\theta(q_\phi(x)) \to 0 \text{ (for decoder)} \quad (8)$$

For the equivariant encoder and decoder, we differently propose the single forward process by the encoder and decoder objective functions compared to previous work (Yang et al., 2022).

**Equivariance Loss for Encoder and Decoder**    To be the equivariant function between the input and latent vector space, the mapping function $q_\phi(\cdot)$ must satisfy Equation 7. Therefore, we directly induce an equivariant encoder between input and latent space with MSE loss ($\mathcal{L}_{ee}$). Also, we induce the equivariant decoder ($\mathcal{L}_{de}$) with MSE loss following Equation 8:

$$\mathcal{L}_{equiv} = \mathcal{L}_{ee} + \epsilon \mathcal{L}_{de} = \text{MSE}(q_\phi(x_i^1), g_i^{-1} \circ q_\phi(x_i^2)) + \epsilon \text{MSE}(p_\theta(g_i \circ (q_\phi(x_i^1)), \psi_i * p_\theta(q_\phi(x_i^1))), \quad (9)$$

where $x_i^2 = \psi_i * x_i^1$. During the training, we replace the $p_\theta(q_\phi(x_i^1))$ as a $x_i^1$ because the ELBO term includes the reconstruction error between $p_\theta(q_\phi(x_i^1))$ and $x_i^1$ to be close to zero.

# 4    RELATED WORK

**Disentanglement Learning**    Diverse works for unsupervised disentanglement learning have elaborated in the machine learning field. The VAE based approaches have factorized latent vector dimensions with weighted hyper-parameters or controllable weighted values to penalize Kullback-Leibler divergence (KL divergence) (Higgins et al., 2017; Shao et al., 2020; 2022). Extended works penalize total correlation for factorizing latent vector dimensions with divided KL divergence (Chen et al., 2018) and discriminator (Kim & Mnih, 2018). Differently, we induce disentanglement learning with group equivariant VAE for inductive bias.

**Group Theory Based Approaches for Disentangled Representation**    In recent period, various unsupervised disentanglement learning research proposes different approaches with another definition of disentanglement, which is based on the group theory (Higgins et al., 2018). To learn the equivariant function, Topographic VAE (Keller & Welling, 2021a) proposes the sequentially permuted activations on the latent vector space called shifting temporal coherence, and Groupified VAE (Yang et al., 2022) method proposes that inputs pass the encoder and decoder two times to implement permutation group equivariant VAE models. Also, Commutative Lie Group VAE (CLG-VAE) (Zhu et al., 2021; Mercatali et al., 2022) maps latent vectors into Lie algebra with one-parameter subgroup decomposition for inductive bias to learn the group structure from abstract canonical point to inputs. Differently, we propose the trainable symmetries that are extracted between two samples directly on the latent space while maintaining the equivariance function between input and latent vector space.

**Symmetry Learning with Equivariant Model**    Lie group equivariant CNN (Dehmamy et al., 2021) and (Finzi et al., 2020) construct the In the other literature, several works extract symmetries, which consist of matrices, between two inputs or objects. Miyato et al. (2022) extracts the symmetries between sequential or sequentially augmented inputs by penalizing the transformation of difference of the same time interval. Other work extracts the symmetries by comparing two inputs, in which the differentiated factor is a rotation or translation, and implements symmetries with block diagonal matrices (Bouchacourt et al., 2021). Furthermore, Marchetti et al. (2023) decomposes the class and pose factor simultaneously by invariant and equivariant loss function with weakly supervised learning. The unsupervised learning work (Winter et al., 2022) achieves class invariant and group equivariant function in less constraint condition. Differently, we generally extend the class invariant and group equivariant model in the more complex disparity condition without any knowledge of the factors of datasets.

# 5    EXPERIMENTS

## 5.1    SETTINGS

We implement $\beta$-VAE (Higgins et al., 2017), $\beta$-TCVAE (Chen et al., 2018), Factor-VAE (Kim & Mnih, 2018), control-VAE (Shao et al., 2020), and Commutative Lie Group VAE (CLG-VAE) (Zhu

| 3D Car | FVM | beta VAE | MIG | SAP | DCI | m-FVM$_2$ | m-FVM$_3$ | m-FVM$_4$ |
|---|---|---|---|---|---|---|---|---|
| $\beta$-VAE | 91.83($\pm$4.39) | 100.00($\pm$0.00) | 11.44($\pm$1.07) | 0.63($\pm$0.24) | 27.65($\pm$2.50) | 61.28($\pm$9.40) | - | - |
| $\beta$-TCVAE | 92.32($\pm$3.38) | 100.00($\pm$0.00) | 17.19($\pm$3.06) | 1.13($\pm$0.37) | 33.63($\pm$3.27) | 59.25($\pm$5.63) | - | - |
| Factor-VAE | 93.22($\pm$2.86) | 100.00($\pm$0.00) | 10.84($\pm$0.93) | 1.35($\pm$0.48) | 24.31($\pm$2.30) | 50.43($\pm$10.65) | - | - |
| Control-VAE | 93.86($\pm$5.22) | 100.00($\pm$0.00) | 9.73($\pm$2.24) | 1.14($\pm$0.54) | 25.66($\pm$4.61) | 46.42($\pm$10.34) | - | - |
| CLG-VAE | 91.61($\pm$2.84) | 100.00($\pm$0.00) | 11.62($\pm$1.65) | 1.35($\pm$0.26) | 29.55($\pm$1.93) | 47.75($\pm$5.83) | - | - |
| CFASL | **95.70($\pm$1.90)** | 100.00($\pm$0.00) | **18.58($\pm$1.24)** | **1.43($\pm$0.18)** | 34.81($\pm$3.85) | **62.43($\pm$8.08)** | - | - |

| smallNORB | FVM | beta VAE | MIG | SAP | DCI | m-FVM$_2$ | m-FVM$_3$ | m-FVM$_4$ |
|---|---|---|---|---|---|---|---|---|
| $\beta$-VAE | 60.71($\pm$2.47) | 59.40($\pm$7.72) | 21.60($\pm$0.59) | 11.02($\pm$0.18) | 25.43($\pm$0.48) | 24.41($\pm$3.34) | 15.13($\pm$2.76) | - |
| $\beta$-TCVAE | 59.30($\pm$2.52) | 60.40($\pm$5.48) | 21.64($\pm$0.51) | 11.11($\pm$0.27) | 25.74($\pm$0.29) | 25.71($\pm$3.51) | 15.66($\pm$3.74) | - |
| Factor-VAE | 61.93($\pm$1.90) | 56.40($\pm$1.67) | **22.97($\pm$0.49)** | 11.21($\pm$0.49) | 24.84($\pm$0.72) | 26.43($\pm$3.47) | 17.25($\pm$3.50) | - |
| Control-VAE | 60.63($\pm$2.67) | 61.40($\pm$4.33) | 21.55($\pm$0.53) | 11.18($\pm$0.48) | **25.97($\pm$0.43)** | 24.11($\pm$3.41) | 16.12($\pm$2.53) | - |
| CLG-VAE | 62.27($\pm$1.71) | 62.60($\pm$5.17) | 21.39($\pm$0.67) | 10.71($\pm$0.33) | 22.95($\pm$0.62) | 27.71($\pm$3.45) | 17.16($\pm$3.12) | - |
| CFASL | **62.73($\pm$3.96)** | **63.20($\pm$4.13)** | 22.23($\pm$0.48) | **11.42($\pm$0.48)** | 24.58($\pm$0.51) | **27.96($\pm$3.00)** | **17.37($\pm$2.33)** | - |

| dSprites | FVM | beta VAE | MIG | SAP | DCI | m-FVM$_2$ | m-FVM$_3$ | m-FVM$_4$ |
|---|---|---|---|---|---|---|---|---|
| $\beta$-VAE | 73.54($\pm$6.47) | 83.20($\pm$7.07) | 13.19($\pm$4.48) | 5.69($\pm$1.98) | 21.49($\pm$6.30) | 53.80($\pm$10.29) | 50.13($\pm$11.98) | 48.02($\pm$8.98) |
| $\beta$-TCVAE | 79.19($\pm$5.87) | 89.20($\pm$4.73) | 23.95($\pm$10.13) | 7.20($\pm$0.66) | 35.33($\pm$9.07) | 61.75($\pm$6.71) | 57.82($\pm$5.39) | 63.81($\pm$9.45) |
| Factor-VAE | 78.10($\pm$4.45) | 84.40($\pm$5.55) | 25.74($\pm$10.58) | 6.37($\pm$1.82) | 32.30($\pm$9.47) | 58.39($\pm$5.18) | 51.63($\pm$2.88) | 53.71($\pm$4.22) |
| Control-VAE | 69.64($\pm$7.67) | 82.80($\pm$7.79) | 5.93($\pm$2.78) | 3.89($\pm$1.89) | 12.42($\pm$4.95) | 38.99($\pm$9.31) | 29.00($\pm$10.75) | 19.33($\pm$5.98) |
| CLG-VAE | **82.33($\pm$5.59)** | 86.80($\pm$3.43) | 23.96($\pm$6.08) | 7.07($\pm$0.86) | 31.23($\pm$5.32) | 63.21($\pm$8.13) | 48.68($\pm$9.59) | 51.00($\pm$8.13) |
| CFASL | 82.30($\pm$5.64) | **90.20($\pm$5.53)** | **33.62($\pm$8.18)** | **7.28($\pm$0.63)** | **46.52($\pm$6.18)** | **68.32($\pm$0.13)** | **66.25($\pm$7.36)** | **71.35($\pm$12.08)** |

| 3D Shapes | FVM | beta VAE | MIG | SAP | DCI | m-FVM$_2$ | m-FVM$_3$ | m-FVM$_4$ |
|---|---|---|---|---|---|---|---|---|
| $\beta$-VAE | 84.33($\pm$10.65) | 91.20($\pm$4.92) | 45.80($\pm$21.20) | 8.66($\pm$3.80) | 66.05($\pm$7.44) | 70.26($\pm$6.27) | 61.52($\pm$8.62) | 60.17($\pm$8.48) |
| $\beta$-TCVAE | 86.03($\pm$3.49) | 87.80($\pm$3.34) | 60.02($\pm$10.05) | 5.88($\pm$0.79) | 70.38($\pm$4.63) | 70.20($\pm$4.08) | 63.79($\pm$5.66) | **63.61($\pm$5.90)** |
| Factor-VAE | 79.54($\pm$10.72) | 95.33($\pm$5.01) | 52.68($\pm$22.87) | 6.20($\pm$2.15) | 61.37($\pm$12.46) | 66.93($\pm$17.49) | 63.55($\pm$18.02) | 57.00($\pm$21.36) |
| Control-VAE | 81.03($\pm$11.95) | 95.00($\pm$5.60) | 19.61($\pm$12.53) | 4.76($\pm$2.79) | 55.93($\pm$13.11) | 62.22($\pm$11.35) | 55.83($\pm$13.61) | 51.66($\pm$12.08) |
| CLG-VAE | 83.16($\pm$8.09) | 89.20($\pm$4.92) | 49.72($\pm$16.75) | 6.36($\pm$1.68) | 63.62($\pm$3.80) | 65.13($\pm$5.26) | 58.94($\pm$6.59) | 60.51($\pm$7.62) |
| CFASL | **89.70($\pm$9.65)** | **96.20($\pm$4.85)** | **62.12($\pm$13.38)** | **9.28($\pm$1.92)** | **75.49($\pm$8.29)** | **74.26($\pm$2.82)** | **67.68($\pm$2.67)** | 63.48($\pm$4.12) |

Table 1: Disentanglement scores for single factor change (left 5 metrics) and multi-factor change (m-FVMs) with 10 random seeds.

et al., 2021) for baseline. For common settings to baselines, we set batch size 64, learning rate 1e-4, and random seed from $\{1, 2, \ldots, 10\}$ without weight decay. We train for $3 \times 10^5$ iterations on dSprites smallNORB and 3D Cars, and $5 \times 10^5$ iterations on 3D Shapes. Also, each dataset guarantees the commutativity of transformation. More details for experimental settings are in Appendix C.

## 5.2 QUANTITATIVE ANALYSIS RESULTS AND DISCUSSION

**Disentanglement Performance in Single and Multi-Factor Change** We evaluate four common disentanglement metrics: FVM (Kim & Mnih, 2018), MIG (Chen et al., 2018), SAP (Kumar et al., 2018), and DCI (Eastwood & Williams, 2018), and more details of evaluation settings are in Appendix C. As shown in Table 1, our method gradually improves the disentanglement learning in dSprites, 3D Cars, 3D Shapes, and smallNORB datasets in most metrics. This result also shows that our method positively affects single factor change condition. More details are in Appendix D.1.

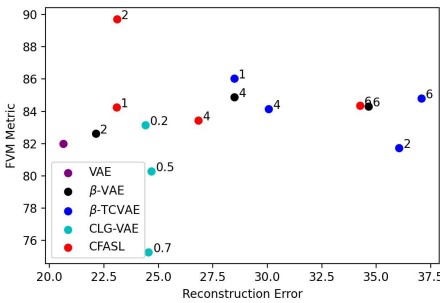

Figure 4: Reconstruction loss vs. Factor VAE metric on 3D Shapes dataset. The numbers next to each plot represent the value of $loss_{\text{rec group}}$ of CLG-VAE and the others are the value of $\beta$ parameter.

To show the quantitative score of the disentanglement in multi-factor change, we evaluate the m-FVM$_k$, where $\max(k)$ is 2, 3, and 4 in 3D Cars, smallNORB, and dSprites datasets respectively. As shown in Table 1, the proposed method shows a statistically significant improvement. It implies that our method has the benefit of disentanglement learning in the multi-factor change condition. We provide additional results in Appendix D.1.

As shown in Figure 4, we show the scatter plot of reconstruction loss against the Factor VAE metric (FVM) on the 3D Shapes dataset. The proposed method CFASL shows a high improvement in the disentanglement performance but a slight cost of reconstruction while the other baselines sacrifice reconstruction.

| | $\mathcal{L}_p$ | $\mathcal{L}_c$ | $\mathcal{L}_{e.}$ | $\mathcal{L}_{pl}$ | $\mathcal{L}_{pd}$ | $\mathcal{L}_s$ | FVM | MIG | SAP | DCI | m-FVM$_2$ |
|---|---|---|---|---|---|---|---|---|---|---|---|
| | ✗ | ✗ | ✗ | ✗ | ✗ | ✗ | 88.19(±4.60) | 6.82(±2.93) | 0.63(±0.33) | 20.45(±3.93) | 42.36(±7.16) |
| | ✗ | ✓ | ✓ | ✓ | ✓ | ✓ | 88.57(±6.68) | 7.18(±2.52) | **1.85**(±1.04) | 18.39(±4.80) | 48.23(±5.51) |
| | ✓ | ✓ | ✗ | ✓ | ✓ | ✓ | 88.56(±7.78) | 7.27(±4.16) | 1.31(±0.70) | 19.58(±4.45) | 42.63(±4.21) |
| $\beta$-VAE | ✓ | ✓ | ✓ | ✗ | ✓ | ✓ | 86.95(±5.96) | 7.11(±3.49) | 1.09(±0.40) | 18.35(±3.32) | 41.90(±7.80) |
| | ✓ | ✓ | ✓ | ✓ | ✗ | ✓ | 85.42(±7.89) | 7.30(±3.73) | 1.15(±0.70) | 21.69(±4.70) | 41.90(±6.07) |
| | ✓ | ✓ | ✓ | ✗ | ✗ | ✗ | 89.34(±5.18) | 9.44(±2.91) | 1.26(±0.40) | **23.14**(±5.51) | 51.37(±9.29) |
| | ✓ | ✓ | ✓ | ✓ | ✓ | ✗ | 90.71(±5.75) | 9.29(±3.74) | 1.07(±0.65) | 22.74(±5.06) | 45.84(±7.71) |
| | ✓ | ✓ | ✓ | ✓ | ✓ | ✓ | **91.91**(±3.45) | **9.51**(±2.74) | 1.42(±0.52) | 20.72(±3.65) | **55.47**(±10.09) |

Table 2: Ablation study for loss functions on 3D-Cars and $\beta$-VAE with 10 random seeds.

**Ablation Study** Table 2 shows the ablation study to evaluate the impact of each component of our method for disentanglement learning. To compare factor-aligned losses (w/o $\mathcal{L}_{pl}$, w/o $\mathcal{L}_{pd}$, w/o $\mathcal{L}_s$, and w/o $\mathcal{L}_{pl} + \mathcal{L}_{pd} + \mathcal{L}_s$), the best of among four cases is the w/o $\mathcal{L}_{pl} + \mathcal{L}_{pd} + \mathcal{L}_s$ and it implies that these losses are interrelated. In the case of w/o $\mathcal{L}_{pl}$, extracting the composite symmetry $g_c$ is difficult, because the role of each section is not unified. Composite symmetry $g_c$ is affected by the second section selection method, which is whether to use the section or not (0 or 1). Therefore, composite with having a different role on elements in the same section struggle with constructing adequate composite symmetry $g_c$. With the similar perspective referred to w/o $\mathcal{L}_{pl}$ case, the coverage of code w/o $\mathcal{L}_{pd}$ is restricted, because there is no guarantee that each section aligns on different factors. In the case of w/o $\mathcal{L}_s$, each section assigns a different role and the elements of each section align on the same factor, w/o $\mathcal{L}_s$ case is better than w/o $\mathcal{L}_{pl}$ and w/o $\mathcal{L}_{pd}$. Also, constructing the symmetries without the equivariant model is meaningless because the model does not satisfy Equation 7- 9. The w/o $\mathcal{L}_{equiv}$ naturally shows the lowest results compared to other cases except w/o $\mathcal{L}_{pd}$ and $\mathcal{L}_{pl}$. Moreover, the w/o $\mathcal{L}_p$ case shows the impact of the second section selection for unsupervised learning. Most of all, each group shows a positive impact on disentanglement compared to the base model ($\beta$-VAE). Integrating all loss functions, the method shows the best performance in most metrics except DCI. The inductive bias for symmetry changes ($\mathcal{L}_{pl} + \mathcal{L}_{pd}$) is less effective than that for composition because the bias is only for symmetry change control without latent dimension matching to a factor. Adding sparsity loss, this issue is resolved and shows the best improvement. More details are in the Appendix D.2.

## 5.3 QUALITATIVE ANALYSIS RESULTS AND DISCUSSION

**Disentanglement in Multi-Factor Change** Previously shown in Fig. 1 is a clear example of the effectiveness of our approach in multi-factor change, because our method forms layers over the shape factor consistently in most x-pos and y-pos factor combinations. Also, the result of the method applied on $\beta$-TCVAE is close to the ideal case, which forms the flat planes parallel to the x-y plane.

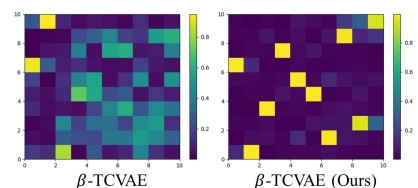

$\beta$-TCVAE     $\beta$-TCVAE (Ours)

Figure 5: Heatmaps of Eigenvectors for latent vector representations.

**Alignment of Factor, Latent Dimensions, and Symmetries** In the principal component analysis of latent vectors shown in Fig. 5, the eigenvectors $V = [v_1, v_2, \ldots, v_{|D|}]$ are close to one-hot vectors compared to the baseline, and the dominating dimension of the one-hot vectors are all different. This result implies that the representation changes are aligned to latent dimensions, and the changes are disentangled by a dimension supposed to work as a factor.

**Generation Quality of Composite Symmetries** To verify the quality of trained composite symmetries, we randomly select a sample pair (red box in Fig. 6a), extract the composite symmetries between both, and obtain other (the bottom right image in a blue box) by applying $g_c$ to the latent vector $z_1$ ($p_\theta(g_c z_1)$) as shown in $1^{st}$ box of Fig. 6a. The bottom left image in a blue box is a generated image from input $x_1$. As shown in Fig. 6b, the model generates the right targets by composite symmetries.

**Disentanglement of Symmetries by Factors** To confirm the distinction of symmetries by the factors, we generate images by sequentially ($p_\theta(z_1), p_\theta(g_1^1 z_1), p_\theta(g_2^1 z_1), \ldots$) applying the symmetry of each factor of the composite symmetry ($\prod_{i,j} g_j^i$) as shown in $2^{nd}$ left box of Fig. 6a. In the second row of Fig. 6c, each symmetry changes only one factor.

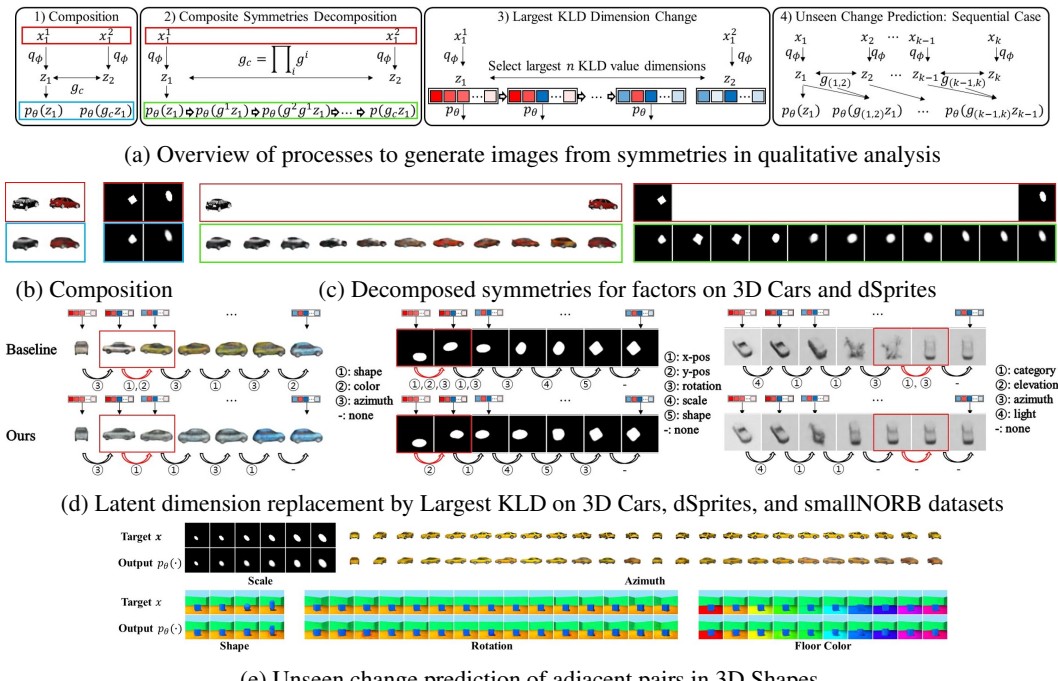

(a) Overview of processes to generate images from symmetries in qualitative analysis

(b) Composition    (c) Decomposed symmetries for factors on 3D Cars and dSprites

(d) Latent dimension replacement by Largest KLD on 3D Cars, dSprites, and smallNORB datasets

(e) Unseen change prediction of adjacent pairs in 3D Shapes

Figure 6: Qualitative Analysis to Generate Images from Latent Vectors in Various Conditions. More details are in the Appendix D.3.

**Disentanglement of Latent Dimensions by Factors**  To analyze the distinction of factors by latent dimensions, we select two random samples, generate latent vectors $z_1$ and $z_2$, and select the largest Kullback-Leibler divergence (KLD) value dimension from their posterior. Then, replacing the dimension value of $z_1$ to the value of $z_2$ one by one sequentially, we generate new images as the process shown in the $3^{rd}$ left box of Fig. 6a. In Fig. 6d, generated images in the red box show that the change of a latent dimension causes multi-factor changes in the baseline model, but our method changes only a factor.

**Unseen Change Prediction in Sequential Data**  We evaluate the quality of unseen change prediction of our method in images sequentially changing a factor. Training with the data, our method randomly selects pairs of images, so the number of pairs composed of adjacent images is rare (less 1 pair among $|\mathbb{X}|/2$ for sequential image dataset $\mathbb{X}$). We set pairs $\{(x_{i-1}, x_i)|1 \leq i \leq ||\mathbb{X}|| - 1\}$ then extract the symmetries between elements of each pair $g_p = \{g_{(1,2)}, g_{(2,3)}, \cdots g_{(k-1,k)}\}$ in inference step, where $g_{(k-1,k)}$ is a symmetry between $z_{k-1}$ and $z_k$. In the last, we generate the sequential outputs as shown in the last box of Fig. 6a. In Fig. 6e, all pairing of adjacent samples is mostly unused in training, but their generated images via trained symmetries of our method are similar. This result implies that our method is strongly regularized for unseen change.

## 6  CONCLUSION

This work tackles the difficulty of disentanglement learning of VAEs in multi-factor change condition. We propose a novel framework to learn composite symmetries from factor-aligned symmetries to directly represent the multi-factor change of a pair of samples. The framework enhances disentanglement by learning an explicit symmetry codebook, injecting three inductive biases on the symmetries aligned to unknown factors, and inducing a group equivariant VAE model. We quantitatively evaluate disentanglement in the condition by a novel metric (m-FVM$_k$) extended from a common metric for a single factor change condition. This method significantly improved in the multi-factor change and gradually improved in the single factor change condition compared to state-of-the-art disentanglement methods of VAEs. This work can be easily plugged into VAEs, and extends disentanglement to more general factor conditions of complex real world datasets.

REPRODUCIBILITY

We provide the code for reproduction and describe the code details in the README.md file in the supplementary materials. Also, we describe the implementation details in the Appendix C.

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

## A    LOSS FUNCTION OF BASELINE

As shown in Table 3, we train the baselines with each objective function.

| VAEs | $\mathcal{L}_{VAE}$ |
|---|---|
| $\beta$-VAE | $\mathbb{E}_{q_\phi(z\|x)} \log p_\theta(x\|z) - \beta \mathcal{D}_{KL}(q_\phi(z\|x)\|\|p(z))$ |
| $\beta$-TCVAE | $\mathbb{E}_{q_\phi(z\|x)} \log p_\theta(x\|z) - \alpha \mathcal{D}_{KL}(q(z,n)\|\|q(z)p(n))$ $-\beta \mathcal{D}_{KL}(q(z)\|\| \prod_j a(z_j)) - \gamma \sum_j \mathcal{D}_{KL}(q(z_j)\|\|p(z_j))$ |
| Factor-VAE | $\frac{1}{N} \sum_i^N [\mathbb{E}_{q(z\|x^i)}[\log p(x^i\|z)] - \mathcal{D}_{KL}(q(z\|x^i)\|\|p(z))]$ $-\gamma \mathcal{D}_{KL}(q(z)\|\| \prod_j (z_j))$ |
| Control-VAE | $\mathbb{E}_{q_\phi(z\|x)} \log p_\theta(x\|z) - \bar{\beta}(t) \mathcal{D}_{KL}(q_\phi(z\|x)\|\|p(z))$ |
| CLG-VAE | $\mathbb{E}_{a(z\|x)q(t\|z)} \log p(x\|z)p(z\|t)$ $-\mathbb{E}_{q(z\|x)} \mathcal{D}_{KL}(q(t\|z)\|\|p(t)) - \mathbb{E}_{q(z\|x)} \log q(z\|x)$ |

Table 3: Objective Function of the VAEs.

## B    PERPENDICULAR AND PARALLEL LOSS RELATIONSHIP

We define parallel loss $\mathcal{L}_p$ to set two vectors in the same section of the symmetries codebook to be parallel: $z - g_j^i \parallel z - g_{j'}^i, z$ then,

$$\boldsymbol{z} - g_j^i \boldsymbol{z} = c(\boldsymbol{z} - g_{j'}^i \boldsymbol{z}) \tag{10}$$

$$\Rightarrow (1-c)\boldsymbol{z} = (g_j^i - cg_{j'}^i)\boldsymbol{z} \tag{11}$$

$$\Rightarrow (1-c)\boldsymbol{I} = g_j^i - cg_{j'}^i \text{ or } [(1-c)I + cg_{j'}^i - g_j^i]\boldsymbol{z} = 0, \tag{12}$$

where $\boldsymbol{I}$ is an identity matrix and constant $c \in \mathbb{R}$. However, all latent $z$ is not eigenvector of $[(1-c)\boldsymbol{I} + cg_{j'}^i - g_j^i]$. Then, we generally define symmetry as:

$$g_{j'}^i = \frac{1}{c} g_j^i + \frac{c-1}{c} \boldsymbol{I}, \tag{13}$$

where $i, j$, and $j'$ are natural number $1 \le i \le |S|$, $1 \le j, j' \le |SS|$, and $k \ne j$. Therefore, all symmetries in the same section are parallel then, any symmetry in the same section is defined by a specific symmetry in the same section.

We define orthogonal loss $\mathcal{L}_o$ between two vectors, which are in different sections, to be orthogonal: $\boldsymbol{z} - g_j^i \boldsymbol{z} \perp \boldsymbol{z} - g_l^k \boldsymbol{z}$, where $i \ne k$, $1 \le i, k \le |S|$, and $1 \le j, l \le |SS|$. By the Equation 13,

$$\boldsymbol{z} - g_j^i \boldsymbol{z} \perp \boldsymbol{z} - g_l^k \boldsymbol{z} \tag{14}$$

$$\Rightarrow (\frac{1}{c_a} g_a^i + \frac{c_a - 1}{c_a} \boldsymbol{I})\boldsymbol{z} - \boldsymbol{z} \perp (\frac{1}{c_b} g_b^k + \frac{c_b - 1}{c_b} \boldsymbol{I})\boldsymbol{z} - \boldsymbol{z} \tag{15}$$

$$\Rightarrow \frac{1}{c_a}(g_a^i \boldsymbol{z} - \boldsymbol{z}) \perp \frac{1}{c_b}(g_b^k \boldsymbol{z} - \boldsymbol{z}), \tag{16}$$

where $c_a$ and $c_b$ are any natural number, and $1 \le a, b \le |SS|$. Therefore, if two vectors from different sections are orthogonal and satisfied with Equation 13, then any pair of vectors from different sections is always orthogonal.

## C    EXPERIMENT DETAILS

**Device**    We set the below settings for all experiments in a single Galaxy 2080Ti GPU for 3D Cars and smallNORB, and a single Galaxy 3090 for dSprites 3D Shapes and CelebA. More details are in README.md file.

**Datasets**    1) The dSprites dataset consists of 737,280 binary $64 \times 64$ images with five independent ground truth factors(number of values), *i.e.* x-position(32), y-position(32), orientation(40), shape(3), and scale(6), Matthey et al. (2017). Any composite transformation of x- and y-position, orientation (2D rotation), scale, and shape is commutative. 2) The 3D Cars dataset consists of 17,568 RGB $64 \times 64 \times 3$ images with three independent ground truth factors: elevations(4), azimuth directions(24), and car models(183) Reed et al. (2015). Any composite transformation of elevations(x-axis 3D rotation), azimuth directions (y-axis 3D rotation), and models are commutative. 3) The smallNORB (LeCun

et al., 2004) dataset consists of total $96 \times 96$ 24,300 grayscale images with four factors, which are category(10), elevation(9), azimuth(18), light(6) and we resize the input as $64 \times 64$. Any composite transformation of elevations(x-axis 3D rotation), azimuth (y-axis 3D rotation), light, and category is commutative. 4) The 3D Shapes dataset consists of 480,000 RGB $64 \times 64 \times 3$ images with six independent ground truth factors: orientation(15), shape(4), floor color(10), scale(8), object color(10), and wall color(10) Burgess & Kim (2018). 5) The CelebA dataset Liu et al. (2015) consists of 202,599 images, and we crop the center $128 \times 128$ area and then, resize to $64 \times 64$ images.

**Evaluation Settings** We set *prune_dims.threshold* as 0.06, 100 samples to evaluate global empirical variance in each dimension, and run it a total of 800 times to estimate the FVM score introduced in Kim & Mnih (2018). For the other metrics, we follow default values introduced in Michlo (2021), training and evaluation 10,000 and 5,000 times with 64 mini-batches, respectively Cao et al. (2022).

**Model Hyper-parameter Tuning** We implement $\beta$-VAE (Higgins et al., 2017), $\beta$-TCVAE (Chen et al., 2018), control-VAE (Shao et al., 2020), Commutative Lie Group VAE (CLG-VAE) (Zhu et al., 2021), and Groupified-VAE (G-VAE) (Yang et al., 2022) for baseline. For common settings to baselines, we set batch size 64, learning rate 1e-4, and random seed from $\{1, 2, \ldots, 10\}$ without weight decay. We train for $3 \times 10^5$ iterations on dSprites smallNORB and 3D Cars, $6 \times 10^5$ iterations on 3D Shapes, and $10^6$ iterations on CelebA. We set hyper-parameter $\beta \in \{1.0, 2.0, 4.0, 6.0\}$ for $\beta$-VAE and $\beta$-TCVAE, fix the $\alpha, \gamma$ for $\beta$-TCVAE as 1 (Chen et al., 2018). We follow the ControlVAE settings (Shao et al., 2020), the desired value $C \in \{10.0, 12.0, 14.0, 16.0\}$, and fix the $K_p = 0.01$, $K_i = 0.001$. For CLG-VAE, we also follow the settings (Zhu et al., 2021) as $\lambda_{hessian} = 40.0$, $\lambda_{decomp} = 20.0$, $p = 0.2$, and balancing parameter of $loss_{\text{rec group}} \in \{0.1, 0.2, 0.5, 0.7\}$. For G-VAE, we follow the official settings (Yang et al., 2022) with $\beta$-TCVAE ($\beta \in \{10, 20, 30\}$), because applying this method to the $\beta$-TCVAE model usually shows higher performance than other models (Yang et al., 2022). Then we select the best case of models. We run the proposed model on the $\beta$-VAE and $\beta$-TCVAE because these methods have no inductive bias to symmetries. We set the same hyper-parameters of baselines with $\epsilon \in \{0.1, 0.01\}$, threshold $\in \{0.2, 0.5\}$, $|S| = |SS| = |D|$, where $|D|$ is a latent vector dimension. More details for experimental settings.

## C.1 BEST MODELS FOR QUANTITATIVE ANALYSIS

In this section, we show how we pick the best model among hyper-parameter tuning results. As shown in Table 4-6, we choose the best model on each datasets.

| $\beta$ | beta VAE | FVM | MIG | SAP | DCI |
|---|---|---|---|---|---|
| 1.0 | 78.80($\pm$6.61) | 65.13($\pm$12.78) | 4.62($\pm$3.21) | 2.67($\pm$1.52) | 9.22($\pm$3.05) |
| 2.0 | 81.00($\pm$7.62) | 64.78($\pm$10.02) | 6.34($\pm$3.66) | 3.37($\pm$1.70) | 10.95($\pm$4.42) |
| 4.0 | **82.67**($\pm$7.28) | **73.54**($\pm$6.47) | **13.19**($\pm$4.48) | **5.69**($\pm$1.98) | **21.49**($\pm$6.30) |
| 6.0 | 74.80($\pm$10.46) | 63.20($\pm$6.76) | 8.35($\pm$2.95) | 2.43($\pm$1.27) | 13.45($\pm$5.07) |

(a) $\beta$-VAE

| $\beta$ | beta VAE | FVM | MIG | SAP | DCI |
|---|---|---|---|---|---|
| 1.0 | 77.20($\pm$8.01) | 65.46($\pm$8.79) | 4.32($\pm$1.46) | 2.41($\pm$1.30) | 9.34($\pm$1.23) |
| 2.0 | 78.20($\pm$9.59) | 70.68($\pm$11.16) | 11.74($\pm$8.51) | 3.84($\pm$2.83) | 16.80($\pm$11.20) |
| 4.0 | 87.40($\pm$4.72) | 78.18($\pm$7.31) | 19.47($\pm$6.61) | 6.32($\pm$1.70) | 30.05($\pm$8.57) |
| 6.0 | **89.20**($\pm$4.73) | **79.19**($\pm$5.87) | **23.95**($\pm$10.13) | **7.20**($\pm$0.66) | **35.33**($\pm$9.07) |

(b) $\beta$-TCVAE

| $C$ | beta VAE | FVM | MIG | SAP | DCI |
|---|---|---|---|---|---|
| 10.0 | **82.80**($\pm$ 7.79) | **69.64**($\pm$7.67) | **5.93**($\pm$2.78) | **3.89**($\pm$1.89) | **12.42**($\pm$4.95) |
| 12.0 | 75.20($\pm$5.43) | 68.00($\pm$8.67) | 5.10($\pm$2.24) | 2.49($\pm$1.50) | 9.82($\pm$3.69) |
| 14.0 | 73.60($\pm$9.03) | 61.58($\pm$7.87) | 4.53($\pm$2.60) | 2.11($\pm$1.67) | 9.30($\pm$1.89) |
| 16.0 | 76.20($\pm$8.14) | 63.28($\pm$7.98) | 4.09($\pm$2.00) | 2.08($\pm$1.37) | 8.91($\pm$1.88) |

(c) Control-VAE

| $loss_{\text{rec group}}$ | beta VAE | FVM | MIG | SAP | DCI |
|---|---|---|---|---|---|
| 0.1 | 86.80($\pm$3.43) | **82.33**($\pm$5.59) | **23.96**($\pm$6.08) | **7.07**($\pm$0.86) | **31.23**($\pm$5.32) |
| 0.2 | **88.20**($\pm$4.57) | 82.88($\pm$3.55) | 20.39($\pm$6.31) | 6.82($\pm$1.80) | 28.28($\pm$7.09) |
| 0.5 | **88.20**($\pm$5.53) | 81.05($\pm$7.51) | 20.63($\pm$6.64) | 6.49($\pm$1.98) | 27.45($\pm$6.07) |
| 0.7 | 88.00($\pm$4.81) | 79.93($\pm$8.16) | 18.95($\pm$6.86) | 6.94($\pm$1.19) | 27.27($\pm$6.76) |

(d) Commutative Lie Group VAE

Table 4: Baselines hyper-parameter tuning results on dSprites dataset with 10 random seeds.

| $\beta$ | FVM | MIG | SAP | DCI | m-fvm$_2$ |
|---|---|---|---|---|---|
| 1.0 | 88.19(±4.60) | 6.82(±2.93) | 0.63(±0.33) | 20.45(±3.93) | 42.36(±7.16) |
| 2.0 | 88.51(±5.44) | 10.00(±3.84) | **0.79**(±0.38) | 28.78(±7.28) | 50.98(±8.33) |
| 4.0 | 90.95(±4.01) | **12.76**(±1.19) | 0.61(±0.36) | **30.70**(±3.06) | 55.76(±9.97) |
| 6.0 | **91.83**(±4.39) | 11.44(±1.07) | 0.63(±0.24) | 27.65(±2.50) | **61.28**(±9.40) |

(a) $\beta$-VAE

| $\beta$ | FVM | MIG | SAP | DCI | m-fvm$_2$ |
|---|---|---|---|---|---|
| 1.0 | 89.85(±7.17) | 7.27(±3.94) | 0.71(±0.40) | 21.21(±6.26) | 41.99(±3.09) |
| 2.0 | 91.29(±3.95) | 11.62(±3.70) | 0.79(±0.27) | 30.60(±5.23) | 54.87(±3.20) |
| 4.0 | **92.70**(±3.41) | **17.31**(±2.91) | 1.07(±0.36) | 33.19(±3.38) | 59.12(±2.20) |
| 6.0 | 92.32(±3.38) | 17.20(±3.06) | **1.13**(±0.36) | **33.63**(±3.27) | **59.25**(±5.63) |

(b) $\beta$-TCVAE

| $C$ | FVM | MIG | SAP | DCI | m-fvm$_2$ |
|---|---|---|---|---|---|
| 10.0 | **93.86**(±5.12) | **9.73**(±2.24) | **1.14**(±0.54) | **25.66**(±4.61) | **46.42**(±10.34) |
| 12.0 | 91.43(±5.32) | 8.65(±3.59) | 11.28(±0.70) | 21.05(±3.93) | 46.06(±11.20) |
| 14.0 | 88.09(±5.46) | 6.11(±3.46) | 1.05(±0.57) | 22.09(±4.34) | 45.87(±10.84) |
| 16.0 | 89.65(±6.87) | 8.12(±3.71) | 0.71(±0.36) | 20.89(±6.30) | 45.77(±10.23) |

(c) Control-VAE

| $loss_{\text{rec group}}$ | FVM | MIG | SAP | DCI | m-fvm$_2$ |
|---|---|---|---|---|---|
| 0.1 | **91.64**(±3.91) | 10.68(±3.18) | 1.22(±0.47) | **31.24**(±5.42) | 45.74(±8.68) |
| 0.2 | 91.18(±3.18) | 11.45(±1.12) | 1.06(±0.25) | 31.09(±4.15) | 48.12(±6.04) |
| 0.5 | 90.19(±3.46) | 10.90(±1.53) | 1.51(±0.30) | 30.68(±3.01) | **49.53**(±8.44) |
| 0.7 | 91.61(±2.84) | **11.62**(±1.65) | **1.35**(±2.61) | 29.55(±1.93) | 47.75(±5.83) |

(d) Commutative Lie Group VAE

Table 5: Baselines hyper-parameter tuning results on 3D Cars dataset with 10 random seeds.

| $\beta$ | beta VAE | FVM | MIG | SAP | DCI |
|---|---|---|---|---|---|
| 1.0 | **59.40**(±7.72) | **60.71**(±2.47) | **21.61**(±0.59) | **11.02**(±0.18) | **25.43**(±0.48) |
| 2.0 | 56.80(±7.90) | 54.69(±2.96) | 19.97(±0.31) | 10.45(±0.24) | 21.15(±0.47) |
| 4.0 | 52.40(±7.65) | 55.19(±1.73) | 19.14(±0.49) | 9.67(±0.24) | 20.54(±0.41) |
| 6.0 | 52.67(±7.28) | 53.42(±1.54) | 18.05(±0.27) | 10.10(±0.28) | 21.03(±0.27) |

(a) $\beta$-VAE

| $\beta$ | beta VAE | FVM | MIG | SAP | DCI |
|---|---|---|---|---|---|
| 1.0 | **60.40**(±5.48) | 59.30(±2.52) | 21.64(±0.51) | **11.11**(±0.27) | **25.74**(±0.29) |
| 2.0 | 56.60(±9.24) | **59.48**(±2.14) | **21.72**(±0.44) | 11.08(±0.35) | 23.74(±0.33) |
| 4.0 | 58.00(±6.86) | 56.40(±1.55) | 21.50(±0.62) | 10.98(±0.35) | 22.29(±0.73) |
| 6.0 | 56.00(±8.17) | 55.46(±1.42) | 21.49(±0.52) | 10.50(±0.25) | 20.24(±0.51) |

(b) $\beta$-TCVAE

| $C$ | beta VAE | FVM | MIG | SAP | DCI |
|---|---|---|---|---|---|
| 10.0 | 59.80(±5.77) | 60.34(±2.58) | 21.53(±0.33) | 10.91(±0.37) | 25.55(±0.49) |
| 12.0 | 60.20(±10.60) | **61.00**(±1.86) | 21.39(±0.41) | **11.25**(±0.32) | 25.71(±0.37) |
| 14.0 | **61.40**(±4.33) | 60.63(±2.67) | **21.55**(±0.53) | 11.18(±0.48) | **25.97**(±0.43) |
| 16.0 | 60.20(±7.69) | 60.50(±2.89) | 21.72(±0.31) | 11.30(±0.41) | 25.60(±0.33) |

(c) Control-VAE

| $loss_{\text{rec group}}$ | beta VAE | FVM | MIG | SAP | DCI |
|---|---|---|---|---|---|
| 0.1 | 59.20(±5.75) | 59.54(±1.64) | 20.61(±0.41) | 10.93(±0.36) | **23.77**(±0.60) |
| 0.2 | 63.40(±9.14) | 59.74(±1.60) | 20.87(±0.36) | **10.80**(±0.47) | 23.59(±0.63) |
| 0.5 | **64.20**(±8.24) | 61.28(±1.68) | 21.20(±0.53) | 10.58(±0.36) | 22.88(±0.52) |
| 0.7 | 62.60(±5.17) | **62.26**(±1.71) | **21.39**(±0.67) | 10.71(±0.33) | 22.95(±0.62) |

(d) Commutative Lie Group VAE

Table 6: Baselines hyper-parameter tuning results on smallNORB dataset with 10 random seeds.

| $p$-value | FVM | MIG | SAP | DCI |
|---|---|---|---|---|
| dSprites | **0.011** | **0.005** | **0.016** | **0.001** |
| 3D Cars | **0.006** | **0.000** | 0.97 | **0.003** |
| smallNORB | **0.000** | **0.002** | **0.000** | 1.000 |

Table 7: $p$-value estimation on each datasets.

| 3D Shapes | $\beta$-VAE | $\beta$-TCVAE | Factor-VAE | Control-VAE | CLG-VAE | OURS |
|---|---|---|---|---|---|---|
| m-FVM$_5$ | 80.26($\pm$3.78) | 79.21($\pm$5.87) | 76.69($\pm$5.08) | 73.31($\pm$6.54) | 73.61($\pm$4.22) | **83.03**($\pm$2.73) |

Table 8: m-FVMs results.

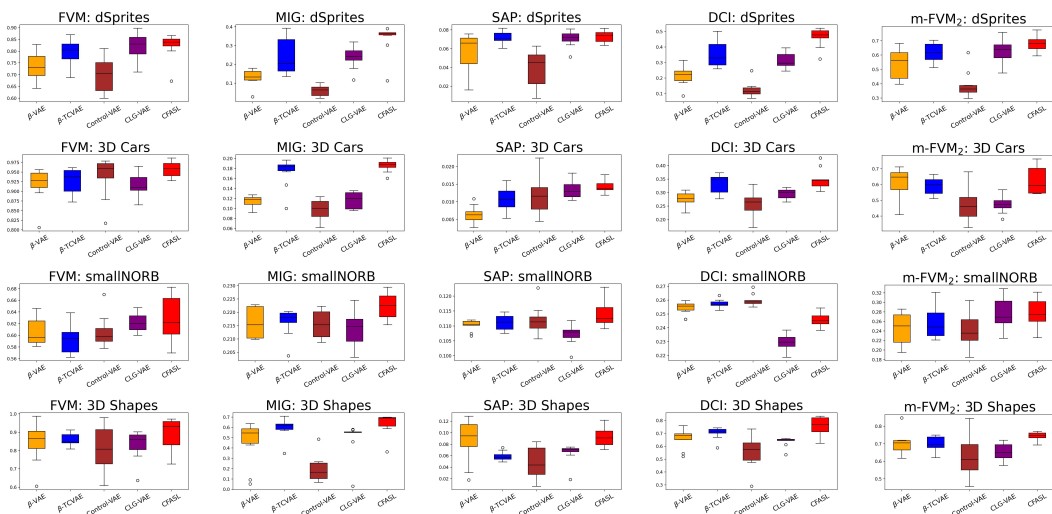

Figure 7: Disentanglement Scores with box plots.

# D    ADDITIONAL EXPERIMENT

## D.1    DISENTANGLEMENT PERFORMANCE

**Statistically Significant Improvements**    As shown in Figure 7, our model significantly improves disentanglement learning.

**3D Shapes**    As shown in Table 8, CFASL also shows an advantage on multi-factor change.

## D.2    ABLATION STUDIES

**How Commutative Lie Group Improves Disetanglement Learning?**    The Lie group is not commutative, however most factors of the used datasets are commutative. For example, 3D Shapes dataset factors consist of the azimuth (x-axis), yaw (z-axis), coloring, scale, and shape. Their 3D rotations are all commutative. Also, other composite symmetries as coloring and scale are commutative. Even though we restrict the Lie group to be commutative, our model shows better results than baselines as shown in Table 1.

**Impact of Hyper-Parameter tuning**    We operate a grid search of the hyper-parameter $\epsilon$. As shown in Figure 8a, the Kullback-Leibler divergence convergences to the highest value, when $\epsilon$ is large ($\epsilon = 1.0$) and it shows less stable results. It implies that the CFASL with larger $\epsilon$ struggles with disentanglement learning, and is shown in Tabel 9a. Also, the $\mathcal{L}_{ee}$ in Figure 8b is larger than other cases, which implies that the model struggles with extracting adequate composite symmetry because its encoder is far from the equivariant model and it is also shown in Table 9a. Even though $\epsilon = 0.01$ case shows the lowest value in the most loss, $\mathcal{L}_{de}$ in Figure 8e is higher than others and it also implies the model struggles with learning symmetries, as shown in Table 9a because the model does not close to the equivariant model compare to $\epsilon = 0.1$ case.

| $\epsilon$ | FVM | beta VAE | MIG | SAP | DCI |
|---|---|---|---|---|---|
| 0.01 | 76.98($\pm$8.63) | 87.33($\pm$7.87) | 29.68($\pm$11.38) | 6.96($\pm$1.16) | 41.28($\pm$11.93) |
| 0.1 | **82.21**($\pm$1.34) | **90.33**($\pm$5.85) | **34.79**($\pm$3.26) | **7.45**($\pm$0.61) | **48.07**($\pm$5.62) |
| 1.0 | 76.77($\pm$7.05) | 78.33($\pm$13.88) | 22.42($\pm$11.14) | 6.02($\pm$0.48) | 38.87($\pm$7.83) |

(a) Hyper-parameter tuning with 6 random seeds.

| 3D Cars | $|\mathcal{G}|$=100 | $|\mathcal{G}|$=10 |
|---|---|---|
| FVM | **95.70**($\pm$1.90) | 48.63($\pm$24.55) |
| MIG | **18.58**($\pm$1.24) | 2.99($\pm$6.04) |
| SAP | **1.43**($\pm$0.18) | 0.29($\pm$0.34) |
| DCI | **34.81**($\pm$3.85) | 6.12($\pm$10.44) |
| FVM$_2$ | **62.43**($\pm$8.08) | 37.94($\pm$10.01) |

(b) Codebook size impact

Table 9: Table

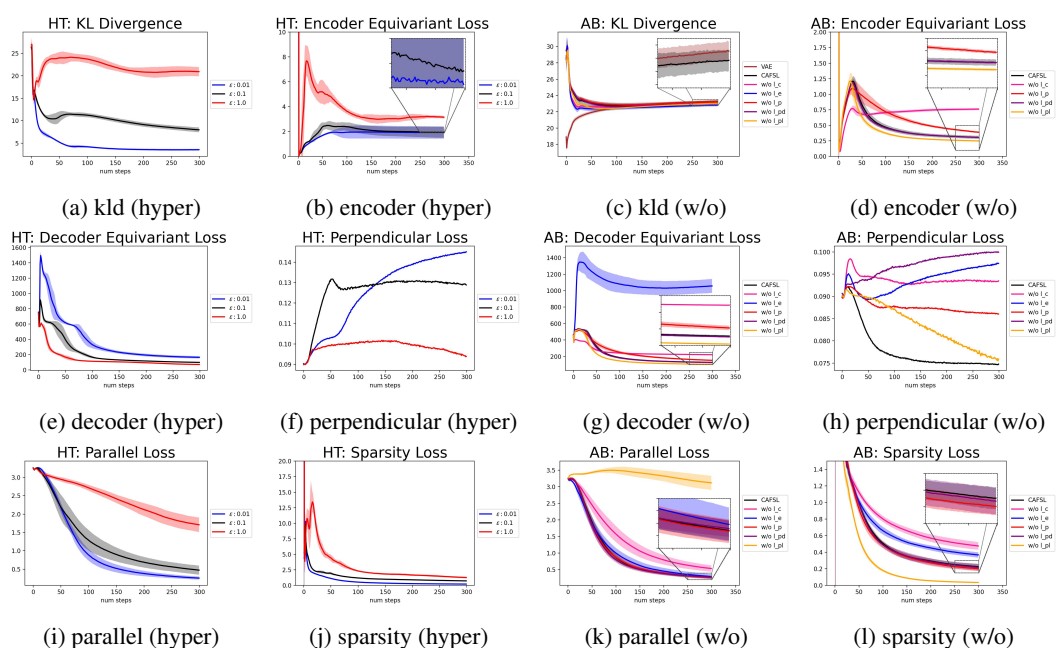

(a) kld (hyper)    (b) encoder (hyper)    (c) kld (w/o)    (d) encoder (w/o)

(e) decoder (hyper)    (f) perpendicular (hyper)    (g) decoder (w/o)    (h) perpendicular (w/o)

(i) parallel (hyper)    (j) sparsity (hyper)    (k) parallel (w/o)    (l) sparsity (w/o)

Figure 8: Loss curves: 1) HT: hyper-parameter tuning ($\epsilon \in \{0.01, 0.1, 1.0\}$) with $\beta$-TCVAE based CFASL. 2) AB: ablation study with $\beta$-VAE based CFASL.

**Impact of Factor-Aligned Symmetry Size**   We set the codebook size as 100, and 10 to validate the robustness of our method. In Table 9b, the larger size shows better results than the smaller one, and is more stable by showing a low standard deviation.

**Impact of Commutative Loss on Computational Complexity**   As shown in Table 10, our methods reduce the composite symmetries computation. Matrix exponential is based on the Taylor series and it needs high computation cost though its approximation is lighter than the Taylor

| 3D Cars | $\mathcal{L}_c$ | without $\mathcal{L}_c$ |
|---|---|---|
| | x**4.63** | x1.00 |

Table 10: Complexity.

series. We need one matrix exponential computation for composite symmetries with commutative loss, in contrast, the other case needs the number of symmetry codebook elements $|S| \cdot |SS|$ for the matrix exponential and also $|S| \cdot |SS| - 1$ time matrix multiplication.

**Comparison of Plug-in Methods**   To compare plug-in methods, we evaluate common disentanglement metrics on G-VAE Shakerinava et al. (2022) and apply both methods to $\beta$-TCVAE. As shown in Table 11, our method shows statistically significant improvements in disentanglement learning although $\beta$ hyper-parameter of CFASL is smaller than G-VAE. As shown in Table 7, we estimate the

| Datasets | FVM | | MIG | | SAP | | DCI | |
|---|---|---|---|---|---|---|---|---|
| | G-VAE | CFASL | G-VAE | CFASL | G-VAE | CFASL | G-VAE | CFASL |
| dSprites | 69.75($\pm$13.66) | **82.30**($\pm$5.64) | 21.09($\pm$9.20) | **33.62**($\pm$8.18) | 5.45($\pm$2.25) | **7.28**($\pm$0.63) | 31.08($\pm$10.87) | **46.52**($\pm$6.18) |
| 3D Car | 92.34($\pm$2.96) | **95.70**($\pm$1.90) | 11.95($\pm$2.16) | **18.58**($\pm$1.24) | **2.10**($\pm$0.96) | 1.43($\pm$0.18) | 26.91($\pm$6.24) | **34.81**($\pm$3.85) |
| smallNROB | 46.64($\pm$1.45) | **61.15**($\pm$4.23) | 20.66($\pm$1.22) | **22.23**($\pm$0.48) | 10.37($\pm$0.51) | **11.12**($\pm$0.48) | **27.77**($\pm$0.68) | 24.59($\pm$0.51) |

Table 11: Comparison of disentanglement scores of plug-in methods in single factor change.

*p*-value over common disentanglement metrics on each dataset. Most values show that improvements in disentanglement learning are statistically significant.

### D.3 ADDITIONAL QUALITATIVE ANALYSIS (BASELINE VS. CFASL)

Fig. 9-10 show the qualitative results on 3D Cars introduced in Fig. 6a. Fig. 11-12, and Fig. 15 show the dSprites and smallNORB dataset results respectively. Additionally, we describe Fig. 13-14, and Fig. 16 results over 3D Shapes, and CelebA datasets respectively. We randomly sample the images in all cases.

**3D Cars** As shown in Figure 10c, CFASL shows better results than the baseline. In the $1^{st}$ and $2^{nd}$ rows, the baseline changes shape and color factor when a single dimension value is changed, but ours clearly disentangle the representations. Also in the $3^{rd}$ row, the baseline struggles with separating color and azimuth but CFASL successfully separates the color and azimuth factors.

- $1^{st}$ row: our model disentangles the *shape* and *color* factors when the $2^{nd}$ dimension value is changed.
- $2^{nd}$ row: ours disentangles *shape* and *color* factors when the $1^{st}$ dimension value is changed.
- $4^{th}$ row: ours disentangles the *color*, and *azimuth* factors when the $2^{nd}$ dimension value is changed.

**dSprites** As shown in Figure 12c, the CFASL shows better results than the baseline. The CFASL significantly improves the disentanglement learning as shown in the $4^{th}$ and $5^{th}$ rows. The baseline shows the multi-factor changes during a single dimension value is changed, while ours disentangles all factors.

- $1^{st}$ row: ours disentangles *the x- and y-pos* factor when the $2^{nd}$ dimension value is changed.
- $2^{nd}$ row: ours disentangles the *rotation* and *scale* factor when the $2^{nd}$ dimension value is changed.
- $3^{rd}$ row: ours disentangles the *x- and y-pos*, and *rotation* factor when the $1^{st}$ and $2^{nd}$ dimension values are changed.
- $4^{th}$ row: ours disentangles the *all factors* when the $1^{st}$ and $2^{nd}$ dimension values are changed.

**3D Shapes** As shown in Figure 14c, the CFASL shows better results than the baseline. In the $1^{st}$, $3^{rd}$, and $5^{th}$ rows, our model clearly disentangles the factors while the baseline struggles with disentangling multi-factors. Even though our model does not clearly disentangle the factors, compared to the baseline, which is too poor for disentanglement learning, ours improves the performance.

- $1^{st}$ row: our model disentangles the *object color* and *floor color* factor when the $2^{nd}$ and $3^{rd}$ dimension values are changed.
- $2^{nd}$ row: ours disentangles *shape* factor in $1^{st}$ dimension, and *object color* and *floor color* factors at the $4^{th}$ dimension value are changed.
- $3^{rd}$ row: ours disentangles the *object color* and *floor color* factor when the $3^{rd}$ dimension value is changed.
- $4^{th}$ row: ours disentangles the *scale*, *object color*, *wall color*, and *floor color* factor when the $2^{nd}$ and $3^{rd}$ dimension values are changed.
- $5^{th}$ row: ours disentangles the *shape*, *object color*, and *floor color* factor when the $1^{st}$ and $2^{nd}$ dimension values are changed.

**smallNORB** Even though our model does not clearly disentangle the multi-factor changes, ours shows better results than the baseline as shown in Figure 15c.

- $1^{st}$ row: our model disentangles the *category* and *light* factor when the $2^{nd}$ dimension value is changed.
- $3^{rd}$ row: ours disentangles *category* factor and *azimuth* factors when the $5^{th}$ dimension value is changed.

**celebA** As shown in Figure 16c, the CFASL shows better results than the baseline. Our model clearly disentangles the multi-factor change while the baseline shows poor disentanglement performance. Furthermore, CFASL learns the sunglasses factor that is not observed in the baseline results. It implies that our model effectively disentangles the multi-factors in the same condition and is much more effective in the complex dataset compared to the baseline.

- $1^{st}$ row: our model disentangles the *background*, *skin color*, and *face shape* factor when the $2^{nd}$.

- $2^{nd}$ row: ours disentangles *azimuth*, *hair length*, and *hair color* factors from $2^{nd}$ to $4^{th}$ dimension values are changed. Furthermore, our method newly learns the ***sunglasses*** factor that is not observed in baseline results.

- $3^{rd}$ row: ours disentangles the *forehead*, *azimuth*, and *background* factor when the $2^{nd}$ and $3^{rd}$ dimension values are changed.

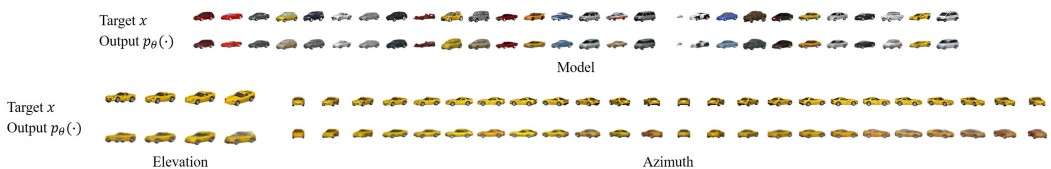

Figure 9: Unseen change prediction in sequential case results on 3D Cars dataset.

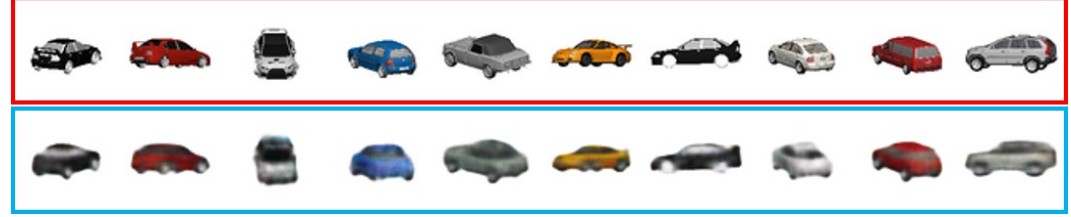

(a) Generation Quality of Composite Symmetries

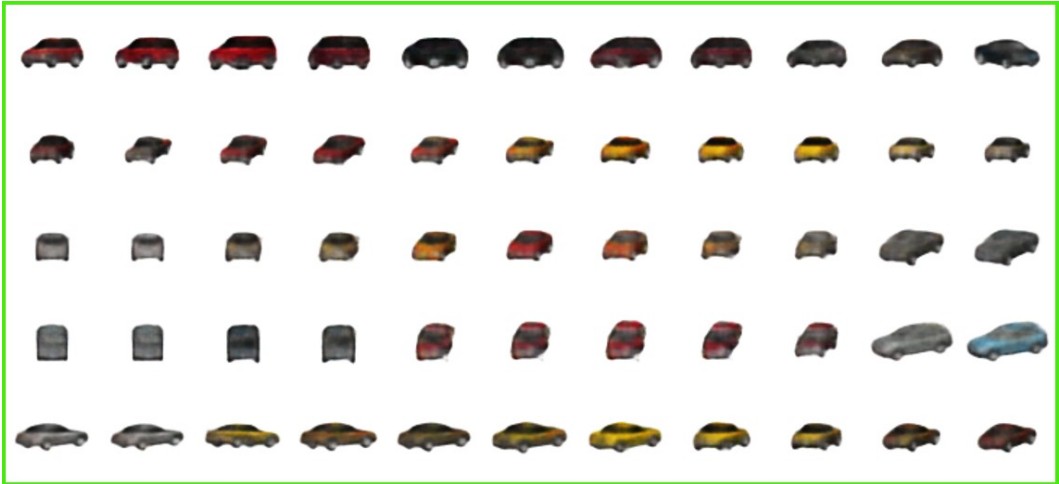

(b) Disentanglement of Symmetries by Factors

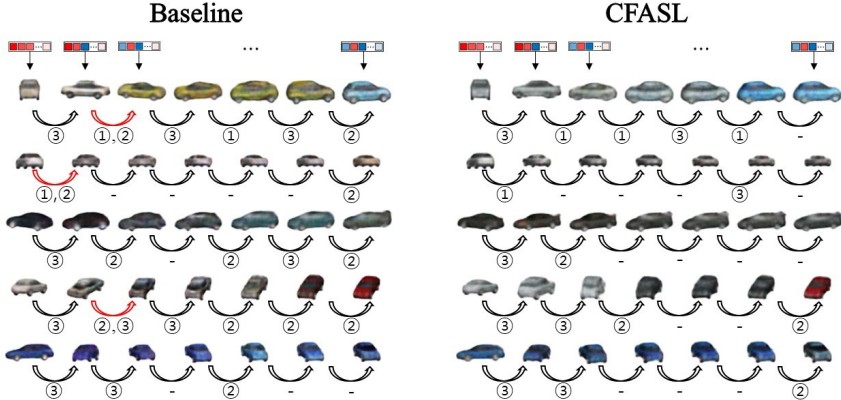

①: shape ②: color ③: azimuth - : none

(c) Disentanglement of latent dimensions by factors

Figure 10: Fig. 10a shows the generation quality of composite symmetries results, Fig. 10b shows the disentanglement of symmetries by factors results, and Fig. 10c shows the disentanglement of latent dimensions by factors results.

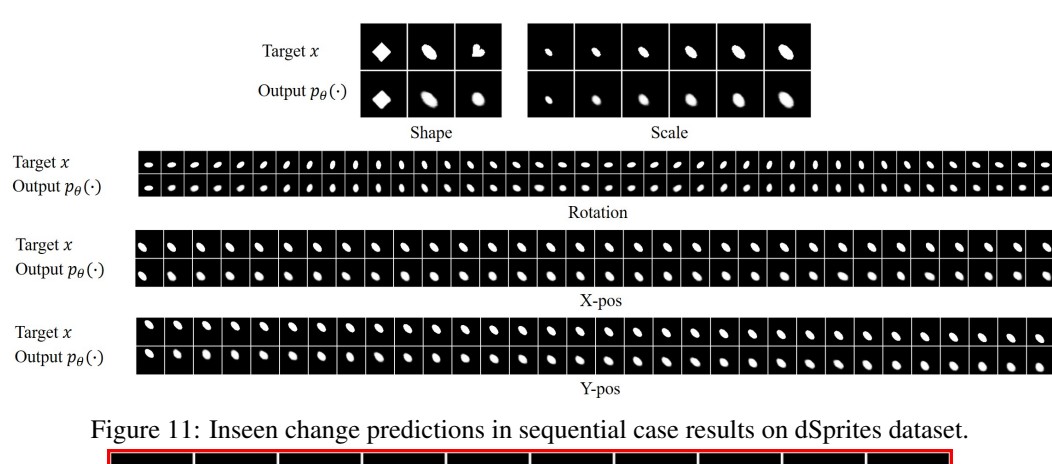

Figure 11: Inseen change predictions in sequential case results on dSprites dataset.

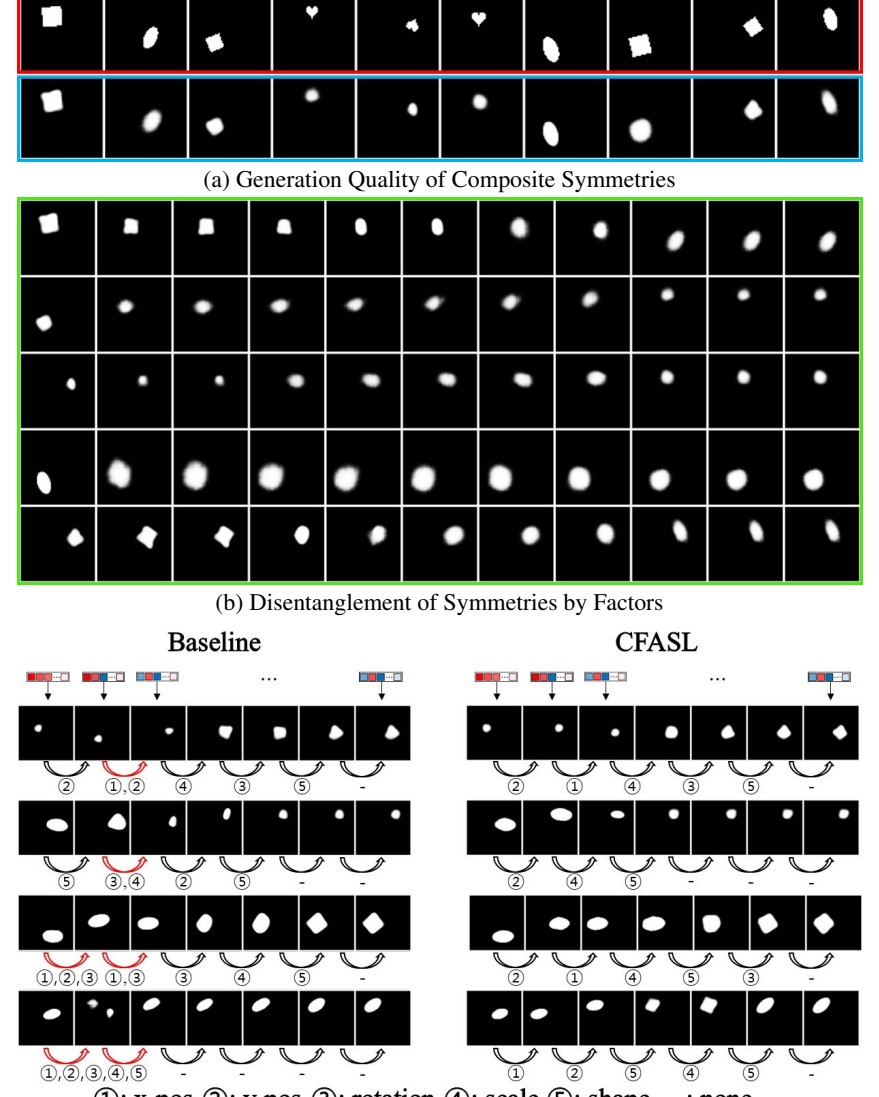

(a) Generation Quality of Composite Symmetries

(b) Disentanglement of Symmetries by Factors

①: x-pos ②: y-pos ③: rotation ④: scale ⑤: shape  -  : none

(c) Disentanglement of latent dimensions by factors

Figure 12: Fig. 12a shows the generation quality of composite symmetries results, Fig. 12b shows the disentanglement of symmetries by factors results, and Fig. 12c shows the disentanglement of latent dimensions by factors results.

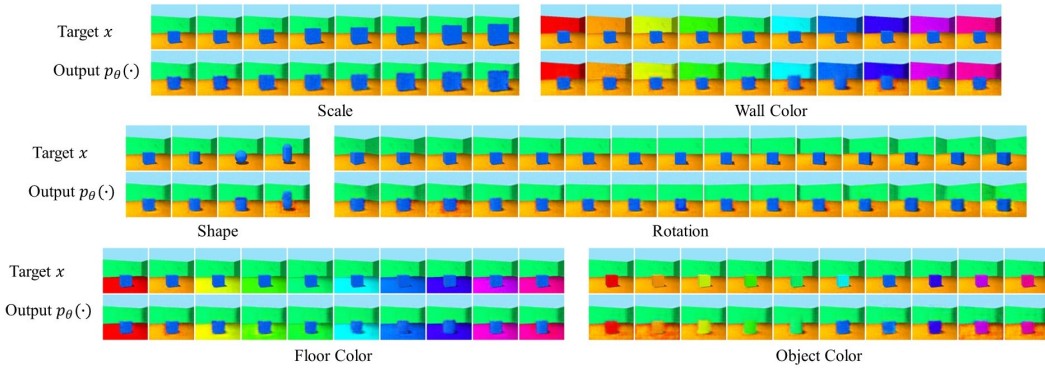

Figure 13: Unseen change predictions in sequential case results on 3D Shapes dataset.

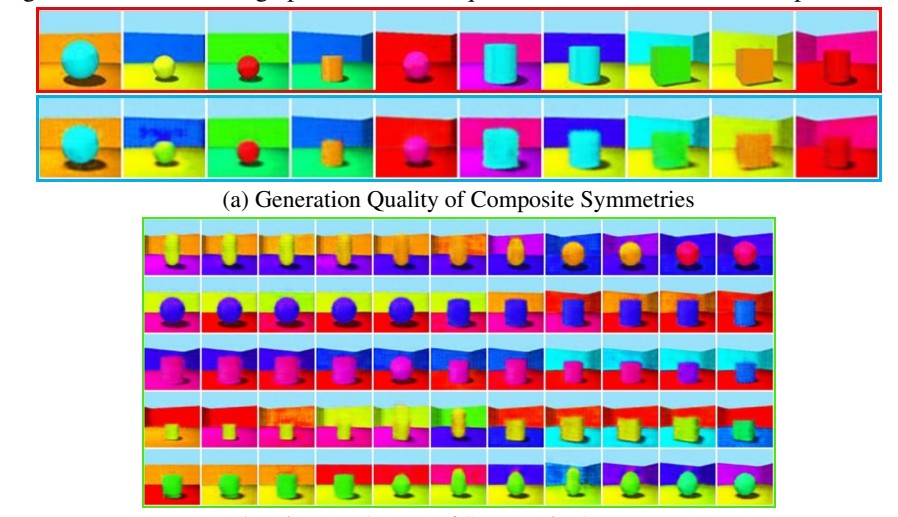

(a) Generation Quality of Composite Symmetries

(b) Disentanglement of Symmetries by Factors

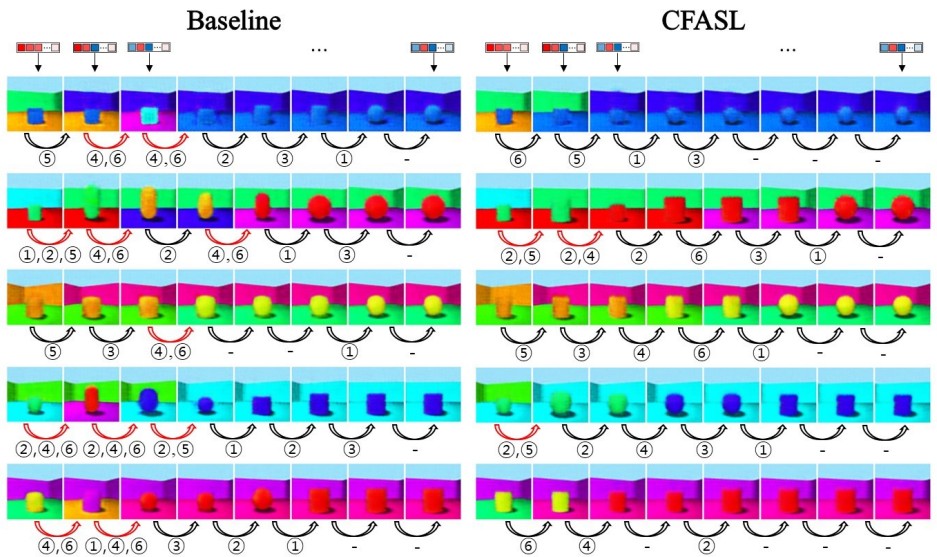

①: shape ②: scale ③: rotation ④: object color ⑤: wall color ⑥: floor color - : none

(c) Disentanglement of latent dimensions by factors

Figure 14: Fig. 14a shows the generation quality of composite symmetries results, Fig. 14b shows the disentanglement of symmetries by factors results, and Fig. 14c shows the disentanglement of latent dimensions by factors results.

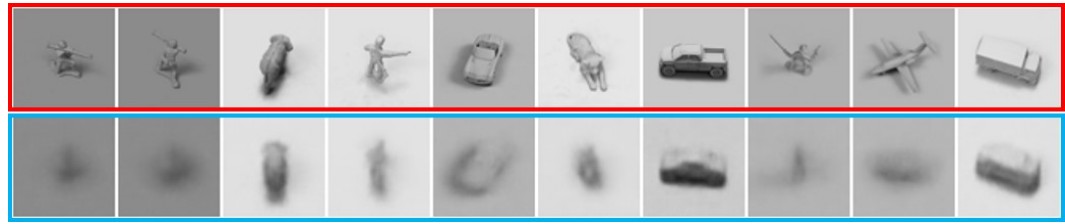

(a) Generation Quality of Composite Symmetries

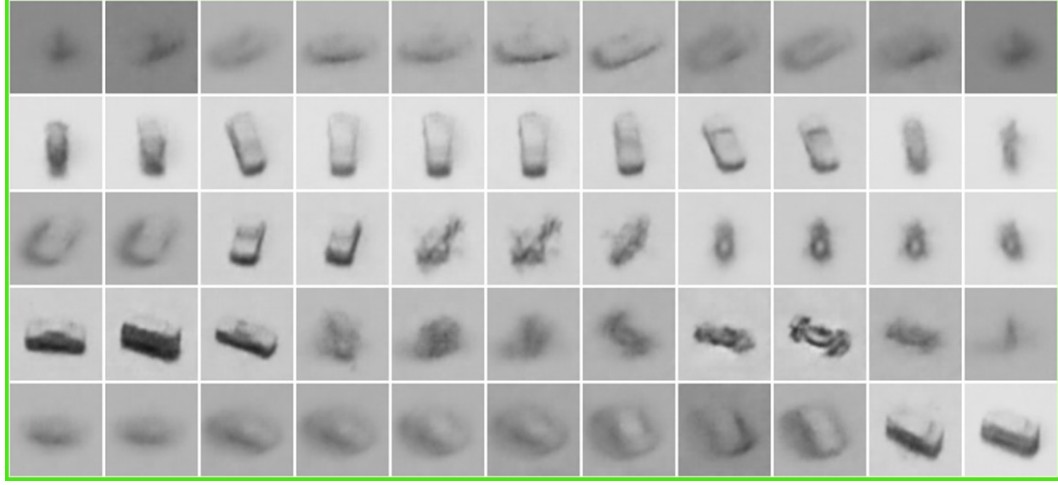

(b) Disentanglement of Symmetries by Factors

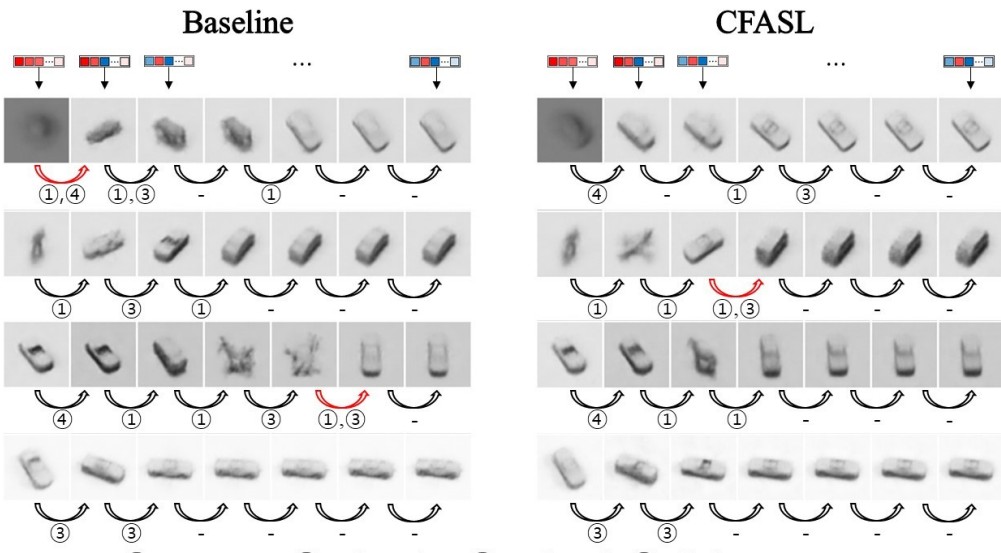

①: category ②: elevation ③: azimuth ④: light - : none

(c) Disentanglement of latent dimensions by factors

Figure 15: Fig. 15a shows the generation quality of composite symmetries results, Fig. 15b shows the disentanglement of symmetries by factors results, and Fig. 15c shows the disentanglement of latent dimensions by factors results.

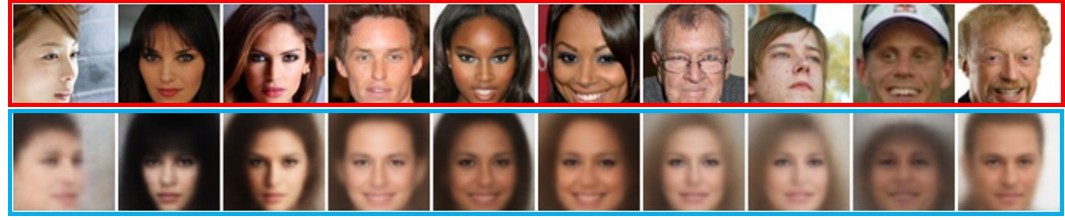

(a) Generation Quality of Composite Symmetries

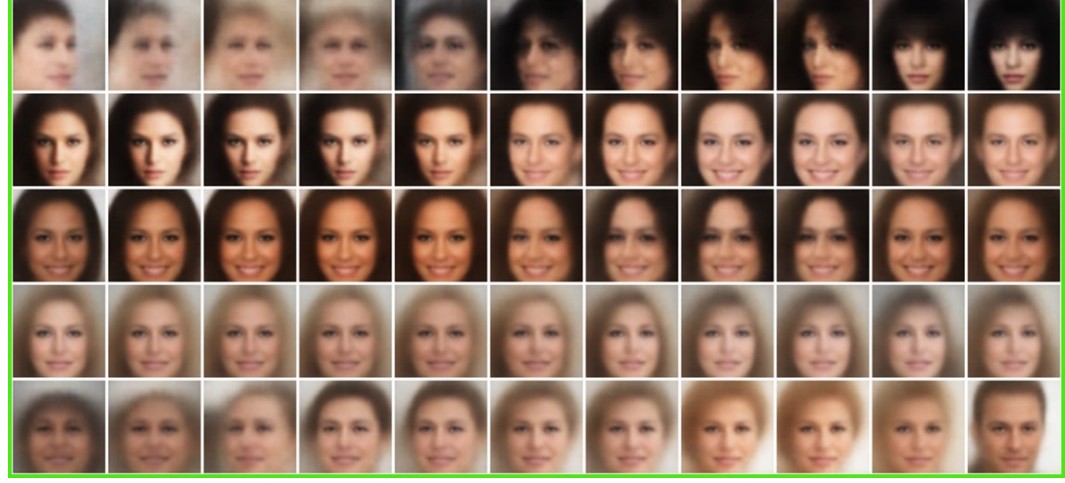

(b) Disentanglement of Symmetries by Factors

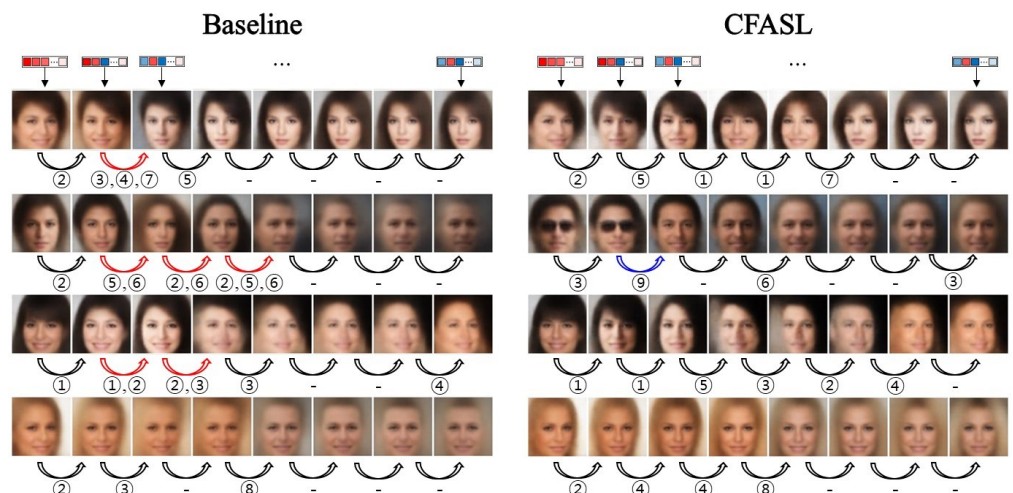

①: forehead ②: azimuth ③: background ④: skin ⑤: hair length
⑥: hair color ⑦: face shape ⑧: brightness ⑨: sunglasses - : none

(c) Disentanglement of latent dimensions by factors, the blue represents the only observed factor in the CFASL.

Figure 16: Fig. 16a shows the generation quality of composite symmetries results, Fig. 16b shows the disentanglement of symmetries by factors results, and Fig. 16c shows the disentanglement of latent dimensions by factors results.

