# OpenReview forum: "CFASL: Composite Factor-Aligned Symmetry Learning for Disentanglement in Variational Autoencoder"
_ICLR.cc/2024/Conference — ICLR 2024 Conference Withdrawn Submission_

### Official Review · Reviewer_5Cgw · 2023-10-21

**Soundness:** 3 good
**Presentation:** 2 fair
**Contribution:** 2 fair
**Rating:** 3
**Confidence:** 5

**Summary:**

The study puts forward a VAE-based approach to acquire disentangled representations without the need for supervision. In this framework, it assumes that diverse data samples exhibit variations across multiple factors, making it particularly well-suited for real-world datasets. The newly proposed technique, referred to as CFASL, introduces a range of unsupervised loss components that serve to instill "inductive biases." These include parallel and perpendicular loss terms, in addition to a sparsity loss designed to encourage alignment along factor axes. The outcomes of this study illustrate the method's superior performance when compared to various other unsupervised disentanglement VAEs, both under single-factor and multi-factor alteration scenarios, across multiple widely used benchmark datasets.

**Strengths:**

1. The paper represents a significant stride in enhancing the practicality of disentanglement techniques within the realm of real image domains. It grapples with a formidable challenge where we cannot presume access to images that solely vary in a singular factor, thereby intensifying the complexity of extracting disentangled representations.

2. The quantitative findings not only exhibit enhancements in the primary focus of this study, which is the alteration of multiple factors, but also in the scenario involving changes in a single factor.

**Weaknesses:**

1. The proposed approach incorporates a diverse array of loss terms within its training objectives, with each term potentially making a distinct contribution. However, this diversity comes at the expense of imposing significant assumptions on the underlying image distribution. While I acknowledge that these assumptions may be justified within the context of the datasets considered in this paper, it's worth noting that some metrics, such as DCI, do not unequivocally demonstrate superiority in the ablation study presented in Table 2.

Nevertheless, I believe that the paper could benefit from a more comprehensive exploration of the limitations stemming from these strong assumptions. It would be valuable for the authors to provide concrete examples where these assumptions result in unintended or adverse outcomes. Even for an unsupervised setting, it remains crucial to take into account the nature of transformations within the image domain. A more explicit discussion of these assumption-related limitations would substantially bolster the significance of the claims advanced in this paper, in my view.

2. The qualitative results exhibit low image quality. While this is common across unsupervised disentanglement methods, it is really challenging to get convinced that better disentanglement is achieved. It would be valuable for the author to consider domain-specific metrics for the evaluation phase e.g. face identity loss, facial expression classification, head pose regression, etc. to assess whether only a specific attribute is altered during the single factor change experiments.

**Questions:**

1. Following the weaknesses mentioned above, could the authors provide concrete examples (other datasets) where the assumptions induced by the loss terms result in unintended or adverse outcomes compared to the baseline beta-VAE?

2. Could the authors please provide the ablation study results of the different loss terms for all datasets considered in the paper (and not only 3D-Cars)?

---

### Official Review · Reviewer_oACj · 2023-10-31

**Soundness:** 3 good
**Presentation:** 1 poor
**Contribution:** 2 fair
**Rating:** 5
**Confidence:** 3

**Summary:**

The authors introduce a new VAE architecture which operates on pairs of inputs and utilizes a set of regularization terms to induce structured disentanglement of the latent space with respect to observed symmetry transformations between examples in these pairs. The authors show that their model indeed achieves higher disentanglement scores than relevant baselines on a variety of datasets with a variety of different metrics. Specifically, the authors target the 'multi-factor change' regime, and demonstrate improved performance in this setting with their newly introduced metric.

**Strengths:**

- The related work is well covered, and the authors position their method well in the literature.
- The proposed combination of losses appears novel to the best of my knowledge, and the use of parallelism and orthogonality losses specifically on latent transformations is an interesting and exciting idea.
- The study of disentanglement with respect to multiple simultaneously changing factors is important and interesting, and the authors make a notable contribution to this direction.
- The results appear promising, and indicate that the model is performing well with respect to the baselines.
- The methodology and extended results in the appendix appear sound. The calculation of P-values in the appendix is very important and appreciated. Furthermore, the use of an ablation study to validate their proposed model is a welcome addition.

**Weaknesses:**

Weaknesses summarized:
- The paper is challenging to read as the english is quite poor and the logical flow of the work is unorganized.
- The method itself is composed of a wide variety of loss terms and the intuition or reasoning for why these terms are necessary is not provided. (Specifically for the parallel and perpendicular losses).

In more detail:

Weakness 1:
There are many typos and poor grammar throughout the paper, with many sentences simply not making much sense. I include a few examples below, but there are many many more and the authors should have someone proof read this work more carefully:
- In the abstract: "We propose ... (CFASL) on VAEs for the extension to [a] general multi-factor change condition without constraint."
- "To implement  group equivariant VAE, Winter et al. (2022); Nasiri & Bepler (2022) achieve the translation and  rotation equivariant VAE"
- "For the equivariant encoder and decoder, we differently propose the single forward process by the  encoder and decoder objective functions compared to previous work (Yang et al., 2022)."
- "Differently, we induce disentanglement learning  with group equivariant VAE for inductive bias."
- 'The unsupervised learning work (Winter et al., 2022) achieves class invariant and group equivariant  function in less constraint condition.'

Weakness 2:
Naming is extremely unclear. For example, what are 'sections' referred to in Section 3.2? How do these differ from factors?

Weakness 3:
Despite appealing to a precise probabilistic generative model as its primary value and distinction from prior work, the model itself could be made significantly more elegant in the context of generative models. For example, the 'factor prediction' mechanism could be integrated as a component of the generative model and inferred with another approximate posterior, as done in prior work (Song et al 2023).

Weakness 4:
The discussion of learning the Lie algebra is quite rushed and the intuition for why the large set of different loss terms should be incorporated is largely missing.

[1] (Song et al. 2023) https://arxiv.org/pdf/2309.13167.pdf

**Questions:**

Question 1:
The point that prior work with autoencoders does not extend to VAE's does not make much sense to me. Specifically the quote: "Furthermore, the methods on autoencoder are not directly applicable to VAEs, because  of the large difference to VAE in probabilistic interpretation". Can the authors provide further details to reinforce this claim?

Question 2:
Given there are so many loss terms for this model, it is likely that it will be computationally expensive to estimate the correct weightings for each of these terms in a hyperparamter search. Can the authors speak to how this was done in their case and how expensive it was?

Question 3:
One of the main selling points for this paper was the ability to extend disentanglement methods to 'multi-factor' change. However, for the experiments, the authors consider datasets which guarantee commutativity of transformations. Theoretically then, is there a reason why we should expect the other baseline models to not be able to handle this multi factor change? For example, it seems the axis aligned disentangled representations of the beta-vae should be able to compose multiple transformations simply by jointly changing multiple latent dimensions. Is this not the case?

---

### Official Review · Reviewer_A4b1 · 2023-11-01

**Soundness:** 2 fair
**Presentation:** 2 fair
**Contribution:** 2 fair
**Rating:** 5
**Confidence:** 3

**Summary:**

Following the Variational Auto Encoder (VAE) framework, this paper proposes an extension of the single factor (change condition) disentanglement learning method, which they call as Composite Factor-Aligned Symmetry Learning (CFASL). The main idea and/or the assumption is certain scenarios such as the composite/complex symmetries (where certain mathematical transformational relationships exist) can be better captured by utilizing explicit symmetrical relationship information, if provided as additional input to the VAE learning framework.

As a part of the learning scheme, to facilitate this required piece of information, the proposed method explicitly inputs pairwise symmetrical relationship (and corresponding transformation) information. The expectation is the model, if learned in this fashion, should generate better representative samples from within those transformational subspace/domains.

To better explain and evaluate the scenario, some new metrics such as m-FVMk (extension of a common metric for a single factor change condition evaluation) have been proposed. They have compared their method with some state-of-the-art methods and on nine benchmark datasets; reported results are found to be promising.

**Strengths:**

The following items seem to have some originality: (i) learning from explicit pairwise transformations, (ii) a network architecture to learn the codebook of symmetries for (i),  (iii) some associated metrics supporting (i) and (ii), and (iv) imposing group equivariant encoder-decoder into the learning framework.

Overall, the paper is well written.  Mathematical derivations of different components seem to be sufficient. The proposed method has been tested on a number of benchmarks (both quantitative and qualitative analysis), and reported results are found to be promising. In addition, the ablation study of different loss functions may have added some extra points.

In terms of quality, I would rate the work as "moderate".

**Weaknesses:**

In this work, one of the important missing part is the proper probabilistic derivation of the methodology, the core of the VAE framework. Or it may be due to the way the paper/work has been presented. To me, it's not sufficient to connect to the VAE world. It is suggested the authors clarify this important aspect with necessary derivations.

For certain items/results, the authors claim statistical significance performance (section 5.2, and appendix D); however, without sufficient details of their significance tests. It is suggested authors include details of these statistical tests.

As the authors have implemented the benchmark approaches (section 5) by themselves, we may require additional details for a fair companion of their results.

The paper/research may have some significance, and it would be beneficial if the source code could be released.

**Questions:**

It is suggested the authors clarify the probabilistic derivation of the approach and make a proper connection to the VAE basics.

It is suggested authors include details of these statistical tests.

As the authors have implemented the benchmark approaches (section 5) by themselves, I suggest authors provide further details and release code if possible.

---

### Official Review · Reviewer_DbMo · 2023-11-02

**Soundness:** 2 fair
**Presentation:** 2 fair
**Contribution:** 2 fair
**Rating:** 3
**Confidence:** 4

**Summary:**

The manuscript aims to improve existing methods of unsupervised disentangled representations learning.  Inspired by the symmetry group action approach from (Higgins et al 2018,2022), authors suggest several additions for the conventional beta-VAE  method, resulting  in the form of seven supplementary loss terms.

**Strengths:**

The article is devoted to important subject of disentanglement learning. Authors report improvements over some of existing methods on four simple datasets

**Weaknesses:**

1) Only simple datasets are considered, the method is not tested on standard complex datasets like MPI 3D.

2) Reported improvements of CFASL in all measured metrics are essentially always situated within standard deviations of some other methods.

3) Reconstruction loss is not reported in 3 out of 4 datasets. Upon visual inspection of reported samples, the reconstruction quality is not satisfactory.

4) As reported on Figure 4, on 3DShapes dataset, there is no consistent improvement in FVM metric even at the expense of deteriorating reconstruction quality .

5) There is no theoretic justifications for introduction of so many, seven in total,  additional loss terms.

6) Description of Lie group action is not clear, how the action by psi_i is defined? how the dimensions of Lie groups are chosen?

7) The described group action by matrix multiplications do not preserve the normal distribution, so the group equivariant term is not compatible with the  standard KL term from beta-VAE loss.

8) There is no comparison with most recent disentanglement methods like DAVA, TCWAE.

9) Related work section does not mention many works from vast literature on disentanglement learning, eg Disentangling Adversarial Variational Autoencoder (ICLR 2023).

**Questions:**

Why is the reconstruction quality not reported in three out of four datasets?

Why the method was not tested on standard more complex datasets like MPI3D?